# When Thinking Fails: The Pitfalls of Reasoning for Instruction-Following in LLMs

**Xiaomin Li**[*]
Harvard University

**Zhou Yu**
Amazon

**Zhiwei Zhang**
Amazon

**Xupeng Chen**
NYU

**Ziji Zhang**
Amazon

**Yingying Zhuang**
Amazon

**Narayanan Sadagopan**
Amazon

**Anurag Beniwal**
Amazon

## Abstract

Reasoning-enhanced large language models (RLLMs), whether explicitly trained for reasoning or prompted via chain-of-thought (CoT), have achieved state-of-the-art performance on many complex reasoning tasks. However, we uncover a surprising and previously overlooked phenomenon: explicit CoT reasoning can significantly degrade instruction-following accuracy. Evaluating 20+ models on two benchmarks: IFEval (with simple, rule-verifiable constraints) and ComplexBench (with complex, compositional constraints), we consistently observe performance drops when CoT prompting is applied. Through large-scale case studies and an attention-based analysis, we identify common patterns where reasoning either helps (e.g., with formatting or lexical precision) or hurts (e.g., by neglecting simple constraints or introducing unnecessary content). We propose a metric, *constraint attention*, to quantify model focus during generation and show that CoT reasoning often diverts attention away from instruction-relevant tokens. To mitigate these effects, we introduce and evaluate four strategies: in-context learning, self-reflection, self-selective reasoning, and classifier-selective reasoning. Our results demonstrate that selective reasoning strategies, particularly classifier-selective reasoning, can substantially recover lost performance. To our knowledge, this is the first work to systematically expose reasoning-induced failures in instruction-following and offer practical mitigation strategies.

## 1 Introduction

Reasoning-enhanced large language models (RLLMs) have demonstrated remarkable success across a variety of tasks, including mathematical problem solving, planning, and multi-hop question answering [Guo et al., 2025, Achiam et al., 2023, Grattafiori et al., 2024, Xu et al., 2023, Zhou et al., 2022, Wu et al., 2024, Fu et al., 2022, Qi et al., 2024, Chae et al., 2024]. A central contributor to these advancements is *chain-of-thought* (CoT) prompting [Wei et al., 2022b, Nye et al., 2021, Joshi et al., 2023, Lanham et al., 2023], which explicitly encourages models to reason step-by-step prior to providing an answer. Recent prominent models, such as DeepSeek-R1 [Guo et al., 2025], Claude [Anthropic, 2025], and OpenAI's O-series [OpenAI, 2024], either incorporate CoT-style reasoning in their fine-tuning procedures or explicitly offer reasoning as an inherent capability. While CoT generally improves performance on complex reasoning tasks, it incurs higher computational cost due to longer outputs and increased inference latency. Moreover, its impact on more structured tasks—such as instruction following—remains underexplored. Instruction following, the ability to comply with user-specified constraints, is essential for alignment, safety, and practical usability of

---

[*]Correspondence to: Xiaomin Li (email: xiaominli@g.harvard.edu).

language models [Ouyang et al., 2022, Zhang et al., 2023]. This raises a natural question: **Does explicit reasoning actually help a model follow instructions more accurately?**

In this paper, we answer this question empirically and arrive at a surprising conclusion: reasoning via CoT can *degrade* a model's ability to follow instructions. To investigate this, we evaluate 20+ language models of varying sizes and training paradigms, including general-purpose models (e.g., Llama, Mixtral) and reasoning-tuned models (e.g., Claude 3.7, DeepSeek-R1), on two complementary instruction-following benchmarks: *IFEval* [Zhou et al., 2023] consists of prompts with simple, independently verifiable constraints (e.g., "write at least 400 words" or "mention `AI` three times"). In contrast, *ComplexBench* [Wen et al., 2024] includes instructions formed through compositional logic, combining multiple dependent constraints via operations like chaining, selection, and nesting. Across both datasets, we observe a consistent and often substantial accuracy drop when models are prompted with CoT. This finding is surprising, given that reasoning is typically expected to improve performance on many tasks even more challenging than instruction following.

To understand this phenomenon, we conduct two complementary analyses. First, we perform a large-scale manual case study of samples where reasoning notably affects performance. We find that reasoning helps in two common scenarios: *(1) satisfying formatting or structural requirements*, and *(2) enforcing lexical constraints that override default tendencies*. However, it often hurts performance by: *(1) over-focusing on high-level content and neglecting simple constraints*, or *(2) introducing redundant or well-intentioned content that unintentionally violates constraints*. We provide detailed examples and analyses of each scenario. Second, we investigate the impact of reasoning through an attention-based analysis. We propose a quantitative measure: *constraint attention*, which is based on attention scores directed toward constraint tokens within instructions. Visualizations of attention patterns during response generation consistently demonstrate reduced constraint awareness when CoT prompting is employed, an effect observed across different datasets, models, and layers. This shift in attention might explain, at least partially, why reasoning can diminish instruction adherence.

Following these insights, we propose and evaluate four methods to mitigate the adverse impacts of reasoning on instruction-following accuracy: (1) **In-context learning**, where we identify and correct typical reasoning-induced failures, incorporating these examples into prompts; (2) **Self-reflection**, prompting models to evaluate and adjust their reasoning processes and candidate responses; (3) **Self-selective reasoning**, allowing models to autonomously decide when reasoning is beneficial; and (4) **Classifier-selective reasoning**, where a trained classifier determines when reasoning is necessary. Among these methods, we find that classifier-selective reasoning strategies offer substantial gains across benchmarks, while self-reflection is particularly helpful for larger models and simple instructions. We thoroughly discuss the comparative strengths and limitations of these mitigation strategies in Section 5.

In summary, our main contributions are listed as follows:

- We evaluate 20+ language models on comprehensive instruction-following benchmarks, revealing that explicit **reasoning can negatively impact instruction-following capabilities**[2]. To our knowledge, **we are the first to discover and systematically explore this phenomenon**.

- We provide detailed analyses of reasoning-induced failures, **categorizing scenarios where reasoning either helps or hinders**, and **quantify attention shifts that explain these performance drops**.

- We **propose and rigorously examine multiple mitigation strategies**, including in-context learning, self-reflection, self-selective reasoning, and classifier-selective reasoning, demonstrating their effectiveness and highlighting promising directions for future research.

## 2   Related Work

**Chain-of-Thought and Reasoning LLMs.**   State-of-the-art large language models (LLMs) often leverage explicit reasoning capabilities, exemplified by models such as OpenAI's O-series [OpenAI, 2024], DeepSeek's R1 [Guo et al., 2025], and Anthropic's Claude [Anthropic, 2025]. These models

---

[2]Code and data for reproducing experiments are available at: `https://github.com/amazon-science/when-thinking-fails-RLLM-if-evaluation`.

are trained on datasets incorporating not just direct responses but also explicit reasoning processes known as Chain-of-Thought (CoT) [Wei et al., 2022b, Nye et al., 2021, Joshi et al., 2023, Lanham et al., 2023]. CoT prompting, which encourages step-by-step reasoning, has demonstrated significant success, particularly in domains requiring complex reasoning, such as mathematics [Guo et al., 2025, Achiam et al., 2023, Grattafiori et al., 2024, Xu et al., 2023, Zhou et al., 2022, Wu et al., 2024, Fu et al., 2022, Qi et al., 2024, Chae et al., 2024]. However, CoT can also introduce additional computational costs and may yield limited or no improvement in certain contexts [Kambhampati et al., 2024, Wang et al., 2024, Sprague et al., 2024]. Our study further explores this dual nature of reasoning, highlighting cases where it can negatively impact instruction-following performance.

**Instruction Following.** Instruction-following is essential for aligning language model outputs with user expectations, enabling the models to reliably execute user-specified tasks. Techniques such as instruction tuning (fine-tuning models on extensive datasets of instruction-response pairs) have been instrumental in fostering this capability [Ouyang et al., 2022, Wang et al., 2023, Longpre et al., 2023, Bai et al., 2022, Wei et al., 2022a, Chung et al., 2022, Muennighoff et al., 2023]. This ability is critical for bridging the gap between pre-training objectives and desirable human-aligned behaviors such as helpfulness and relevance [Radford et al., 2019, Brown et al., 2020]. Despite its importance, instruction-following remains challenging, especially when instructions involve complex, multifaceted requirements [Zhang et al., 2023, He et al., 2024, Gudibande et al., 2023, Kung and Peng, 2023, Heo et al., 2024]. To systematically evaluate LLM instruction adherence, benchmarks such as IFEval [Zhou et al., 2023] and ComplexBench [Wen et al., 2024] have been introduced. IFEval focuses on rule-verifiable, straightforward constraints, while ComplexBench assesses models on sophisticated instructions involving nested and dependent constraints. Further details and comprehensive analyses of these benchmarks will be provided in our experimental evaluations (Section 3).

# 3 Experiments

## 3.1 Datasets and Evaluation Metrics

We use two benchmark datasets, **IFEval** and **ComplexBench**, to comprehensively evaluate the instruction-following capabilities of language models:

**IFEval** is a synthetic dataset consisting of 541 prompts, each associated with one to three verifiable constraints drawn from 25 types (e.g., word count, formatting, keyword usage). We adopt instruction-level *loose accuracy*, which allows minor formatting deviations (e.g., markdown wrappers) to avoid penalizing responses for stylistic differences, thereby better reflecting practical robustness.

**ComplexBench** is a manually curated dataset designed to assess models on complex compositional instructions formed through four operations: *And*, *Chain*, *Selection*, and *Nested*. It contains 1,150 instructions and over 5,300 scoring questions, covering 4 constraint types across 19 dimensions (e.g., lexical, semantic, formatting). Evaluation combines rule-based and LLM-based assessments. For our experiments, we translated all scoring rules into English and manually verified them to construct a fully English-compatible version.

**Evaluation Metrics:** For both datasets, we report the proportion of constraints satisfied per instruction. In ComplexBench, dependency logic applies: failure in a prerequisite constraint results in automatic failure of all dependent constraints.

## 3.2 Models

We evaluate a diverse set of models, including both closed-source models (e.g., `Claude3.7-Sonnet`) and reasoning-focused models (e.g., `DeepSeek-R1`, `Qwen-R1-distilled variants`). Our open-source selection spans parameter scales from 1B to 70B. All model inferences use a temperature of 0. Open-source models are run without quantization using 4 **NVIDIA-H100-80GB** GPUs. In addition to single-model CoT vs. non-CoT comparisons, we also evaluate *paired* variants under a controlled setting: base vs. reasoning-enabled counterparts (e.g., Qwen2.5-Instruct vs. Qwen2.5-Math; Qwen3 Base vs. Qwen3 Think; Claude-3.7-Sonnet vs. Claude-3.7-Sonnet-Think; DeepSeek-V3 vs. DeepSeek-R1), summarized in Table 2.

### 3.3 CoT Prompting

We compare model behavior with and without Chain-of-Thought (CoT) reasoning. The CoT prompts instruct models to reason step by step before producing an answer (exact prompt provided in Appendix F). We then assess instruction-following performance based on the model's answer in each setting.

### 3.4 Results

Performance on IFEval and ComplexBench is reported in the first three columns of Table 1, with visual summaries provided in Figure 3. Notably, 13 out of 14 models experience performance degradation on IFEval when CoT prompting is applied, and all models show declines on ComplexBench. For instance, the accuracy of `Llama3-8B-Instruct` drops from 75.2% to 59.0%, a reduction of over 16 percentage points.

We further compare reasoning-enabled models with their corresponding base variants across *nine* pairs, including `Claude3.7-Sonnet` vs. `Claude3.7-Sonnet-Think`, `DeepSeek-V3` vs. `DeepSeek-R1`, two `Qwen2.5-Instruct` vs. `Qwen2.5-Math` pairs (1.5B, 7B), and three `Qwen3` Base vs. Think pairs (4B, 8B, 32B). These results, shown in Table 2, reveal that reasoning variants typically underperform their base counterparts on instruction-following. While these comparisons are not fully controlled—reasoning-tuned models may undergo additional training stages such as supervised fine-tuning or RL/RLHF—the decline remains evident across both benchmarks.

Overall, these findings uncover a surprising and underexplored vulnerability: explicit reasoning, while often helpful in complex tasks, can increase the likelihood of violating instruction constraints, thereby impairing instruction-following reliability.

## 4 Analysis

To better understand when and why reasoning degrades instruction-following, we conduct two analyses: (1) a manual case study examining when CoT helps or hurts constraint satisfaction, and (2) an attention-based analysis investigating how reasoning shifts model focus away from constraints during generation.

### 4.1 Case Study

We manually examined all 541 samples from IFEval and over 1,000 samples from ComplexBench, focusing on cases where CoT affected whether constraints were satisfied. For each case, we analyzed which constraints were impacted and why the reasoning-based answer either improved or degraded performance.

Due to the length of the instructions and responses, we present detailed illustrative examples in Appendix B. For each example, we include outputs from both the base model and the reasoning-enabled model, showing their respective `Answer` and, for CoT, the full `Thinking` process. We also report the number of constraints satisfied and include a case analysis explaining how reasoning helped or harmed performance. Although we examined a large number of examples, the failure and success cases largely fall into four recurring patterns [3], which we summarize below:

**Reasoning Helps ✓**

1. *Formatting and Structural Adherence:* Reasoning improves compliance with structural constraints, such as producing valid JSON, wrapping the output in double quotes, or following markdown syntax.

2. *Lexical and Keyword Precision:* Reasoning enhances adherence to lexical requirements, including inserting rare characters (e.g., the letter q six times), omitting final punctuation, or using exactly 15 capitalized words.

**Reasoning Hurts ✗**

---

[3]Beyond these behavioral patterns, reasoning also increases computational cost due to longer outputs and additional thinking tokens.

| IFEval Results | Baseline | | Mitigation Methods | | | |
|---|---|---|---|---|---|---|
| Model | Original | CoT | FewShot | SelfReflection | Self-Selective Reasoning | Classifier-Selective Reasoning |
| Claude-3.5-Haiku | 86.9 | 79.5 | 80.2 (↑ 0.7) | 81.7 (↑ 2.2) | 80.8 (↑ 1.3) | **85.8** (↑ 6.3) |
| Claude-3.5-Sonnet | 86.5 | 79.5 | 87.4 (↑ 8.0) | **87.6** (↑ 8.1) | 81.9 (↑ 2.4) | 84.8 (↑ 5.4) |
| Claude-3.7-Sonnet | 90.6 | 90.2 | 89.3 (↓ 0.9) | **92.1** (↑ 1.9) | 90.4 (↑ 0.2) | 90.4 (↑ 0.2) |
| DeepSeek-V3 | 85.2 | 84.3 | 85.2 (↑ 0.9) | **88.3** (↑ 4.0) | 85.3 (↑ 1.0) | 84.2 (↓ 0.1) |
| Llama-3.2-1B-Instruct | 49.0 | 40.7 | 13.1 (↓ 27.5) | 34.9 (↓ 5.7) | 45.7 (↑ 5.0) | **47.5** (↑ 6.8) |
| Llama-3.2-3B-Instruct | 70.4 | 61.9 | 37.2 (↓ 24.8) | 64.9 (↑ 3.0) | 61.9 (↓ 0.0) | **68.8** (↑ 6.8) |
| Meta-Llama-3-8B-Instruct | 75.2 | 59.0 | 36.8 (↓ 22.2) | 65.1 (↑ 6.1) | 59.0 (↓ 0.0) | **69.7** (↑ 10.7) |
| Llama-3.1-70B-Instruct | 85.6 | 77.3 | 44.5 (↓ 32.7) | **84.3** (↑ 7.0) | 77.3 (↓ 0.0) | 83.2 (↑ 5.9) |
| Mixtral-8x7B-Instruct | 53.0 | 56.4 | 32.0 (↓ 24.4) | **55.6** (↓ 0.7) | 33.1 (↓ 23.3) | 55.1 (↓ 1.3) |
| Qwen2.5-1.5B-Instruct | 35.9 | 31.6 | 20.1 (↓ 11.5) | 26.6 (↓ 5.0) | 32.7 (↑ 1.1) | **34.8** (↑ 3.2) |
| DeepSeek-R1-Distill-Qwen-1.5B | 16.8 | 13.7 | **22.0** (↑ 8.3) | 20.0 (↑ 6.3) | 13.9 (↑ 0.2) | 17.0 (↑ 3.3) |
| Qwen2.5-7B-Instruct | 63.6 | 57.7 | 63.4 (↑ 5.7) | **74.3** (↑ 16.6) | 59.7 (↑ 2.0) | 68.8 (↑ 11.1) |
| DeepSeek-R1-Distill-Qwen-7B | 30.9 | 25.1 | 30.1 (↑ 5.0) | **36.4** (↑ 11.3) | 26.2 (↑ 1.1) | 30.1 (↑ 5.0) |

| ComplexBench Results | Baseline | | Mitigation Methods | | | |
|---|---|---|---|---|---|---|
| Model | Original | CoT | FewShot | SelfReflection | Self-Selective Reasoning | Classifier-Selective Reasoning |
| Claude-3.5-Haiku | 66.9 | 62.1 | 59.7 (↓ 2.4) | 61.5 (↓ 0.6) | **67.6** (↑ 5.5) | 67.1 (↑ 5.0) |
| Claude-3.5-Sonnet | 67.5 | 66.0 | 66.9 (↑ 0.9) | 65.5 (↓ 1.5) | 67.6 (↑ 1.6) | **68.1** (↑ 2.1) |
| Claude-3.7-Sonnet | 69.8 | 67.0 | 67.4 (↑ 0.4) | 67.8 (↑ 0.8) | **69.6** (↑ 2.5) | 69.4 (↑ 2.4) |
| DeepSeek-V3 | 71.2 | 71.1 | 66.0 (↓ 5.1) | 65.2 (↓ 5.9) | 71.4 (↑ 0.3) | **71.5** (↑ 0.4) |
| Llama-3.2-1B-Instruct | 36.0 | 26.0 | 19.7 (↓ 6.3) | 23.7 (↓ 2.3) | **36.2** (↑ 10.2) | 35.8 (↑ 9.8) |
| Llama-3.2-3B-Instruct | 50.0 | 45.7 | 49.2 (↑ 3.5) | 44.7 (↓ 1.0) | **49.8** (↑ 4.1) | 49.7 (↑ 4.0) |
| Meta-Llama-3-8B-Instruct | 55.8 | 54.9 | 54.7 (↓ 0.2) | 49.1 (↓ 5.8) | 55.4 (↑ 0.5) | **55.8** (↑ 0.9) |
| Llama-3.1-70B-Instruct | 66.8 | 60.2 | 66.8 (↑ 6.6) | 61.9 (↑ 1.7) | 66.7 (↑ 6.5) | **67.8** (↑ 7.6) |
| Mixtral-8x7B-Instruct | 60.4 | 58.3 | 58.2 (↓ 0.1) | 56.5 (↓ 1.7) | 59.6 (↑ 1.3) | **59.6** (↑ 1.3) |
| Qwen2.5-1.5B-Instruct | 44.1 | 38.8 | 38.1 (↓ 0.7) | 31.3 (↓ 7.5) | **44.0** (↑ 5.2) | **44.0** (↑ 5.2) |
| DeepSeek-R1-Distill-Qwen-1.5B | 18.9 | 16.7 | **24.1** (↑ 7.5) | 20.5 (↑ 3.8) | 18.7 (↑ 2.0) | 18.7 (↑ 2.0) |
| Qwen2.5-7B-Instruct | 60.2 | 52.6 | 60.2 (↑ 7.6) | 55.8 (↑ 3.3) | 63.4 (↑ 10.8) | **63.4** (↑ 10.9) |
| DeepSeek-R1-Distill-Qwen-7B | 46.0 | 38.6 | **50.2** (↑ 11.6) | 37.0 (↓ 1.6) | 46.0 (↑ 7.4) | 45.5 (↑ 6.9) |

Table 1: **Instruction-following performance on IFEval and ComplexBench.** Green and red indicate whether *Original* or *CoT* performed better. Each mitigation column reports the accuracy along with its change relative to CoT, using ↑ for improvement and ↓ for decline. **Bold** values highlight the best-performing mitigation method for each model.

3. *Over-Focusing on High-Level Content and Neglecting Simple Constraints:* When multiple constraints are present, reasoning often emphasizes content planning at the expense of simpler mechanical constraints. Common issues include exceeding word count limits, failing to repeat prompts exactly, using capital letters in lowercase-only tasks, or appending unnecessary content after required phrases.

4. *Introducing Unnecessary Content that Violates Constraints:* Reasoning frequently inserts redundant or well-intentioned additions—such as explanations, translations, or emphasis—that break constraints. Typical violations include: inserting English text into "foreign language only" outputs, including commas in "no commas" tasks, appending commentary to quote-only responses, or exceeding limits on capitalized words.

## 4.2 Constraint-Aware Attention Analysis

To understand why reasoning may degrade instruction-following, we analyze whether models pay less attention to constraint-relevant parts of the prompt during response generation. In many failure cases, we observed that models neglect certain constraints, either by overemphasizing content planning or

| Model Pair | IFEval (%) | ComplexBench (%) |
|---|---|---|
| Claude-3.7-Sonnet vs. Claude-3.7-Sonnet-Think | 90.6 / 90.2 | 69.8 / 69.0 |
| DeepSeek-V3 vs. DeepSeek-R1 | 85.2 / 83.3 | 71.2 / 71.1 |
| Qwen2.5-1.5B-Instruct vs. DeepSeek-R1-Distill-Qwen-1.5B | 35.9 / 13.7 | 44.1 / 16.7 |
| Qwen2.5-7B-Instruct vs. DeepSeek-R1-Distill-Qwen-7B | 63.6 / 25.1 | 60.2 / 38.6 |
| Qwen2.5-1.5B-Instruct vs. Qwen2.5-Math-1.5B | 35.9 / 14.9 | 44.1 / 23.4 |
| Qwen2.5-7B-Instruct vs. Qwen2.5-Math-7B | 63.6 / 27.9 | 60.2 / 28.8 |
| Qwen3-4B vs. Qwen3-4B-Think | 85.0 / 69.5 | 63.8 / 59.3 |
| Qwen3-8B vs. Qwen3-8B-Think | 86.8 / 86.3 | 65.6 / 59.8 |
| Qwen3-32B vs. Qwen3-32B-Think | 87.8 / 85.0 | 70.2 / 63.1 |

Table 2: Comparison of reasoning-enabled models and their non-reasoning counterparts across both benchmarks. Each cell reports accuracy on **IFEval** and **ComplexBench**, with green marking the higher-performing model and red the lower-performing one. Results generally show that explicit reasoning either provides negligible gains or causes small performance drops.

introducing irrelevant information. To investigate this phenomenon systematically, we conduct an attention-based analysis that tracks the model's focus on constraint tokens throughout generation.

We define a *constraint-attention* metric to quantify the model's awareness of constraint-relevant tokens. For each instruction, we first use GPT-4o to automatically extract substrings corresponding to each constraint, and then map them to token indices in the prompt. During generation, we compute attention scores directed toward these tokens for both the reasoning and answer segments. Each model is run twice per instruction: (i) **Base run**: Instruction → Answer, and (ii) **Reasoning run** **(CoT)**: Instruction → Think → Answer. We focus our comparison on the *answer segments* of both runs.

**Transformer Attention.** Let the prompt contain $T_0$ tokens $x_{1:T_0}$, and let the model generate $T$ new tokens $y_{1:T}$. At generation step $t$ ($1 \leq t \leq T$), the visible context is $(x_{1:T_0}, y_{1:t-1})$. Let $h_{t-1}^{(l)}$ denote the hidden state of the most recent token at layer $l$, and let $k \in \{1, \ldots, H\}$ index attention heads. The attention components for layer $l$ and head $k$ are computed as:

$$Q_k^{(l,t)} = W_Q^{(l,k)} h_{t-1}^{(l)}, \quad K_k^{(l)} = W_K^{(l,k)} H_{1:T_0+t-2}^{(l)}, \quad V_k^{(l)} = W_V^{(l,k)} H_{1:T_0+t-2}^{(l)},$$

where $h_{t-1}^{(l)}$ is a single vector (current token), and $H_{1:T_0+t-2}^{(l)}$ is the matrix of all prior hidden vectors (prompt and generated context). This leads to scaled attention weights $A_k^{(l,t)} = \text{softmax}\left(\frac{Q_k^{(l,t)} K_k^{(l)\top}}{\sqrt{d_k}}\right)$. Averaging over all heads yields the layer-level attention vector

$$a^{(l,t)} = \frac{1}{H} \sum_{k=1}^{H} A_k^{(l,t)}.$$

**Constraint Attention.** For each textual constraint $c_r$ ($r = 1, \ldots, R$), we collect the prompt token indices it spans, yielding the index subset $C_r \subseteq \{1, \ldots, T_0\}$. Define the full constraint token set as $C = \bigcup_{r=1}^{R} C_r$. Then we can define the *layer–step constraint attention* (averaged attention to constraint tokens at each layer and step) as

$$\alpha^{(l,t)} = \frac{1}{|C|} \sum_{j \in C} a_j^{(l,t)}.$$

Following this, we compute the **layer-averaged constraint attention at step** $t$ as:

$$\bar{\alpha}^{(t)} = \frac{1}{L} \sum_{l=0}^{L-1} \alpha^{(l,t)}, \tag{1}$$

Based on Equation 1, we visualize the trace of constraint-attention during response generation in Figure 1, showing results for both IFEval and ComplexBench using Qwen2.5-1.5B-Instruct.

Additional examples for other models are provided in Appendix C. After reviewing hundreds of samples, we observe a general trend: reasoning flattens the constraint-attention trace. In cases where reasoning degrades performance, constraint attention during the answer phase is generally lower. In contrast, when reasoning improves performance, we often see a bump in attention aligned with the answer segment.

**Quantifying Attention Drop.** To further quantify these observations, we compute the mean constraint attention across the answer phase. Let $\mathcal{A}$ denote the answer token positions. The average constraint attention at layer $l$ is:

$$\bar{\beta}^{(l)} = \frac{1}{|\mathcal{A}|} \sum_{t \in \mathcal{A}} \alpha^{(l,t)} \tag{2}$$

We define the **attention drop** as the difference between base and CoT runs:

$$\Delta\beta = \bar{\beta}_{\text{Base}} - \bar{\beta}_{\text{CoT}}.$$

We find that $\Delta\beta > 0$ for most cases. Figure 2 shows how this drop varies across layers for WIN vs. LOSE cases. On average, the model exhibits larger attention drops in LOSE cases, particularly in early-to-middle layers. This suggests that **lower constraint attention is predictive of reasoning-induced failures**.

**Conclusion.** Our analysis reveals that explicit reasoning often reduces attention to constraint-relevant parts of the prompt. This diminished awareness increases the risk of violating instructions. Thus, while reasoning is intended to improve task performance, it can unintentionally shift model focus away from critical constraints and harm instruction adherence. In addition, we also explored whether longer reasoning tends to degrade performance and found that **reasoning length does not meaningfully correlate with instruction-following effectiveness** (Appendix J).

## 5    Mitigating Reasoning-Induced Failures in Instruction Following

We introduce and evaluate four strategies designed to mitigate the performance degradation caused by explicit reasoning (via CoT) in instruction-following tasks. To the best of our knowledge, this is the first work to systematically identify and address this issue. Hence, in the absence of existing baselines, we directly compare the effectiveness of our proposed methods. Results are presented in Table 1 and Figure 3, which also include performance differences relative to the CoT baseline.

### 5.1    Method 1: Few-Shot In-Context Learning

We apply in-context learning [Brown et al., 2020] by prepending carefully selected few-shot examples to each instruction. These examples are derived from representative failure cases identified in our case study (Section B) and manually revised to fully satisfy all constraints. Each example includes an `Instruction`, along with a corrected `Thinking` and `Answer`. Full examples are provided in Appendix E.

**Results:** As shown in Table 1, this method yields only modest improvements. We attribute the limited gains to several factors. First, due to the token length limit and the substantial size of each example (including prompt, reasoning, and answer), we were only able to include **a limited number of examples**: four examples for IFEval and three for ComplexBench. Second, because examples were **sourced from outputs of certain models, potentially introducing bias**. These constraints likely reduce the effectiveness of the few-shot strategy and limit its generalizability.

### 5.2    Method 2: Self-Reflection

This method performs a two-step process: the model first generates an initial response with thinking, and then performs a second inference where it reflects on its own reasoning and answer. If the model deems the initial response satisfactory, it retains it as the final output; otherwise, it revises the response and outputs the updated version. The prompt used for self-reflection is detailed in Appendix F.

**Results:** As shown in Table 1, self-reflection yields strong improvements on IFEval—enhancing performance in 11 out of 14 models, and achieving the best results among all four methods for

**(a) IFEval – Attention Traces**        **(b) ComplexBench – Attention Traces**

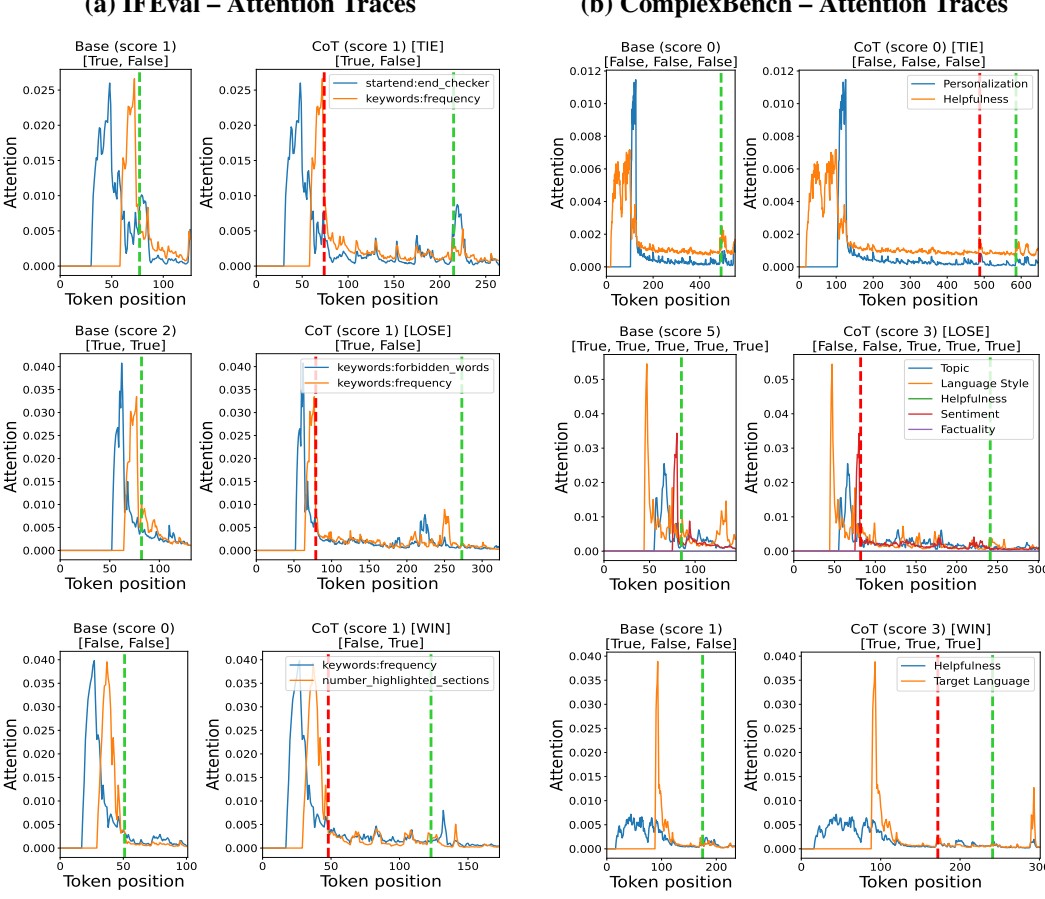

Figure 1: Constraint-attention trace examples for `Qwen2.5-1.5B-Instruct` across both datasets. From top to bottom, the plots show cases where reasoning leads to a TIE, LOSE, and WIN compared to not using reasoning. The vertical red dashed line marks the start of *Thinking*, and the green dashed line marks the start of the *Answer* during response generation. WIN/LOSE/TIE are determined by whether CoT attains a higher, lower, or equal proportion of satisfied constraints than the non-CoT baseline (the "score" is the number of satisfied constraints for that instruction).

7 models. However, we observe notable performance degradation on weaker models such as `Llama-3.2-1B-Instruct` and `Qwen2.5-1.5B-Instruct`. We hypothesize that this is because **self-reflection relies on a model's ability to critique and improve its own outputs, a capability that may be underdeveloped in weaker models**. On ComplexBench, which contains more challenging and compositional instructions, self-reflection proves less effective—leading to performance drops in 10 out of 14 models. This suggests that **self-reflection is more suitable for simpler instruction-following tasks**, and may be counterproductive when applied to complex scenarios. Additionally, a key limitation of this method is its increased computational cost, as it requires two forward passes per query.

### 5.3 Method 3: Self-Selective Reasoning

This method enables the model to decide dynamically whether to perform explicit reasoning. Specifically, we prompt the model to assess, based on the instruction alone, whether reasoning (via CoT) is necessary (see prompt in Appendix F). If the model deems reasoning helpful, it proceeds with step-by-step thinking; otherwise, it directly generates an answer without reasoning.

**Results:** This approach yields moderate gains on IFEval, improving performance in 10 out of 14 models, and shows stronger results on ComplexBench, improving all models and achieving the best

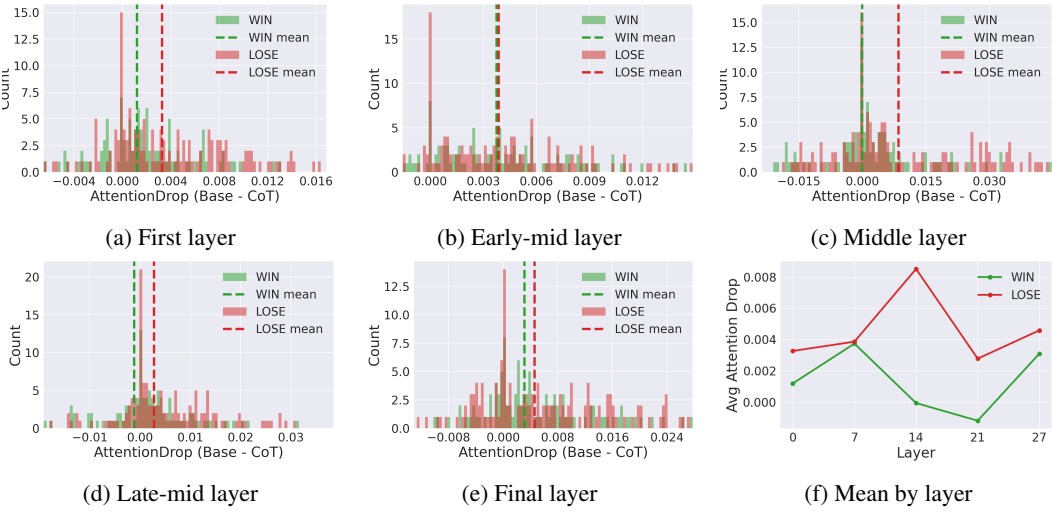

Figure 2: Within IFEval dataset, drop in constraint attention (Base – CoT) for WIN vs. LOSE cases across representative layers of the `Qwen2.5-1.5B-Instruct` model. On average, the drop is more severe in cases when reasoning loses, compared to not using reasoning.

performance among all four methods in 6 models. In Appendix G, we further analyze the model's decision-making behavior by comparing its self-selected reasoning decisions to ground-truth labels based on actual performance differences between CoT and non-CoT responses. We find that while the model tends to have high recall (correctly identifying most cases where reasoning helps) it suffers from low precision, often applying reasoning even when it is not necessary.

## 5.4 Method 4: Classifier-Selective Reasoning

Instead of relying on the model's internal judgment, this method uses an external binary classifier to determine whether CoT reasoning should be applied. For each target model, we train a separate classifier to predict whether using CoT leads to improved instruction-following performance. Labels are assigned on a per-sample basis by comparing the constraint satisfaction scores of CoT and non-CoT responses. Training details are provided in Appendix H.

The classifier is implemented using `Qwen2.5-7B-Instruct` as the backbone and trained for 3 epochs with a learning rate of 1e-5. Both the backbone model and training hyperparameters are selected via grid search. We split the dataset evenly, using 50% of the samples for training and the remaining 50% to evaluate downstream mitigation effectiveness. Validation accuracy typically ranges between 0.75 and 0.92.

**Results:** This method proves highly effective, improving performance for nearly all models on both benchmarks, with only a minor drop for `DeepSeek-V3` on IFEval. **For about half of the models, it achieves the best overall performance among the four strategies.** Nonetheless, its primary drawback is the model-specific training requirement: each model needs its own classifier, which results in additional overhead.

## 5.5 Summary and Recommended Pipeline

Each mitigation strategy exhibits distinct strengths and weaknesses depending on model capacity and instruction complexity. Based on our findings, we propose the following decision pipeline: first, estimate the complexity of the instruction—either through simple heuristics or a trained classifier. For simpler tasks (e.g., IFEval), we recommend **Self-Reflection** or **Classifier-Selective Reasoning**; for more complex or compositional tasks (e.g., ComplexBench), **Self-Selective Reasoning** or **Classifier-Selective Reasoning** is more effective. Overall, **Classifier-Selective Reasoning consistently delivers the best overall performance across both benchmarks**, albeit at the cost of model-specific training.

# 6 Conclusion

One limitation of our study is its exclusive focus on instruction-following tasks. While we suspect that reasoning could similarly degrade performance in other domains, exploring these possibilities is left for future work. In our study, we identified and systematically explored an unexpected phenomenon: explicit reasoning through Chain-of-Thought prompting can negatively impact the instruction-following abilities of large language models. Through extensive evaluation across two comprehensive benchmarks (IFEval and ComplexBench), we demonstrated consistent performance degradation when models employed explicit reasoning. Our detailed analysis, involving manual case studies and attention-based examinations, provided insights into why reasoning negatively affects instruction adherence. We found that reasoning can divert the model's focus from constraint-related tokens, resulting in overlooked or violated instructions.

To address this issue, we introduce and evaluate four mitigation strategies: in-context learning, self-reflection, self-selective reasoning, and classifier-selective reasoning. Our experiments show that selective reasoning, especially classifier-based approaches, can substantially recover lost performance, with classifier-selective reasoning achieving the most consistent improvements across both datasets.

We believe this is the first systematic investigation into reasoning-induced failures in instruction-following tasks. We hope our findings motivate further investigation into reasoning tradeoffs and contribute to building models that reason more selectively and effectively.

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

# Appendix

# A    Experiments: Additional Results

The plots for main results in Table 1 are presented in Figures 3 below.

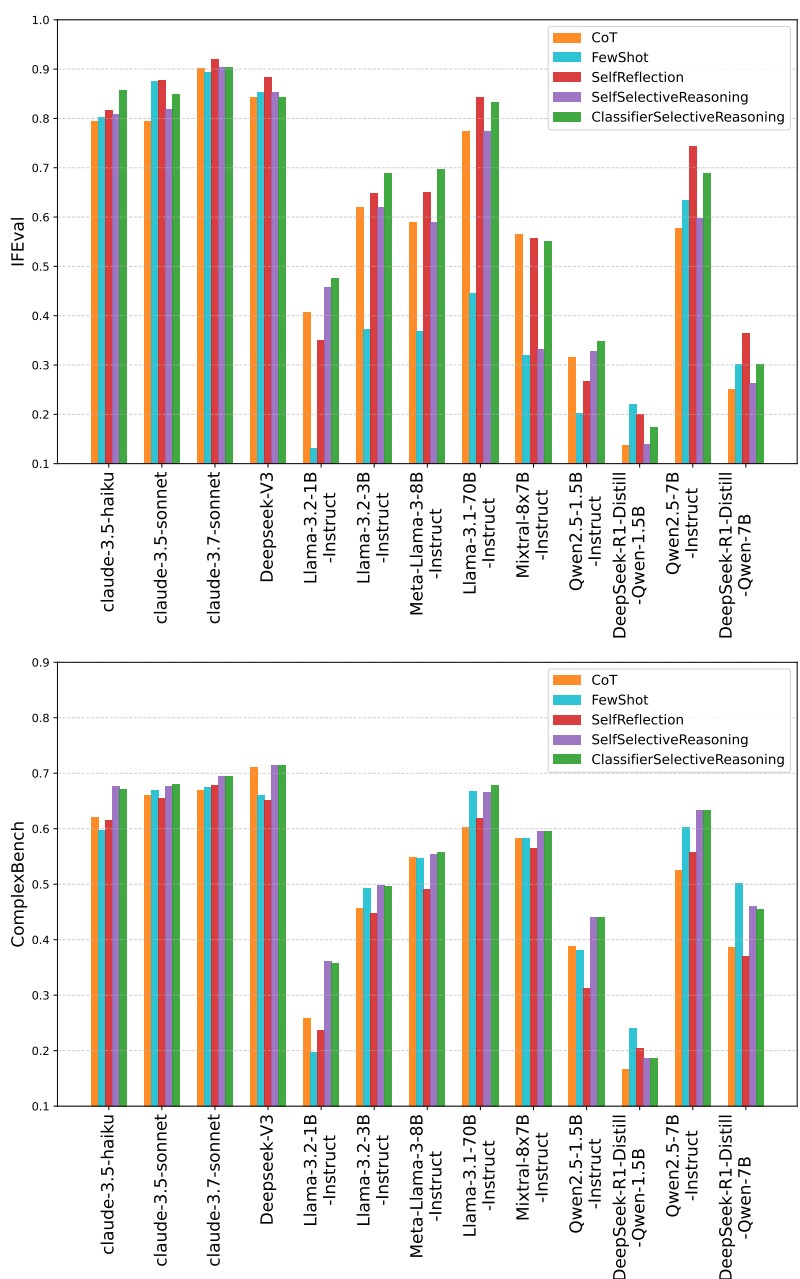

Figure 3: Visualization of instruction-following accuracy across models and methods on IFEval and ComplexBench.

# B    Case Study

We present full examples illustrating the four scenarios discussed in Section 4.1, using outputs from `Claude3.7-Sonnet` and its reasoning-enabled variant `Claude3.7-Sonnet-Think` for direct comparison. To ensure clarity, we focus on examples from IFEval, as prompts and responses in ComplexBench are often too long and complex for a concise presentation. Each example includes

the instruction, its associated constraints, the response generation model, responses with and without reasoning along with their respective constraint satisfaction scores, and a brief analysis explaining how reasoning contributed to the performance difference.

**Reasoning Helps**

1. Reasoning Helps Meet Formatting or Structural Requirements.
2. Reasoning Helps Satisfy Lexical Constraints.

---

**Example 1: Reasoning Helps (Meet Formatting or Structural Requirements)**

**Instruction:** Name a new fashion company that young people might like, and give it a name with multiple meanings. Put the name in double angular brackets, such as `<<name>>`. Repeat the request before responding.

**Constraints:**

- `repeat_prompt` – the model must repeat the instruction before answering
- `title_format` – the name must appear in `<<>>` brackets

**Response Model:** `Claude3.7-Sonnet`

---

**Response without reasoning (constraint satisfaction: [False, True]):**
**Answer:**
I'll help you create a name for a new fashion company that young people might like, with multiple meanings. I'll put the name in double angular brackets as requested.
«Shift»
This name works on multiple levels: 1. "Shift" suggests change and transformation, appealing to young people's desire to express themselves and evolve their style 2. It references the shift key on keyboards, connecting to digital culture 3. In fashion terminology, a shift is also a type of dress, giving it direct industry relevance 4. It implies movement between states or trends, reflecting how young consumers quickly adapt to new styles
The brand could position itself as adaptive, fluid fashion for a generation that's constantly reinventing itself.

---

**Response with reasoning (constraint satisfaction: [True, True]):**
**Thinking:**
I need to name a new fashion company that would appeal to young people. The name should have multiple meanings or interpretations. And I need to present the name in double angular brackets like «name».
The fashion company name should be: - Appealing to young demographics - Have multiple layers of meaning - Be memorable and marketable - Work well for a fashion brand
Some ideas: 1. "Flux" - Represents constant change/movement (fashion trends always changing) and also sounds like "lux" (luxury) 2. "Thread" - Both sewing thread and connecting thread/storyline 3. "Shift" - Changing clothes/shifting perspectives 4. "Alter" - To change clothing/alternative style/altar (elevation) 5. "Cipher" - Code/hidden meaning and also sounds like "cypher" (creative expression in hip-hop culture)
I think "Cipher" works well because: - It has the dual meaning of code/hidden message and connection to creative expression - It sounds modern and intriguing - Young people appreciate brands with depth and cultural connections - It works across various fashion styles
**Answer:**
Name a new fashion company that young people might like, and give it a name with multiple meanings. Put the name in double angular brackets, such as «name».
For a new fashion company appealing to young people, I suggest:
«Thread»
This name works on multiple levels: - Thread literally refers to fabric/clothing materials - It suggests connection and community (threading people together) - In digital culture, it references conversation threads on social media - It hints at continuity and storytelling - threading narratives through fashion - It's concise, modern, and easy to brand

---

**Case Analysis:** The original instruction includes two constraints: repeating the prompt and formatting the name in `<<>>` brackets. Without reasoning, the model skipped repeating the prompt, failing the first constraint. With reasoning, the model:

- Broke down the task and repeated the instruction (satisfying constraint 1)
- Explicitly evaluated multiple name candidates against the constraints
- Chose a final name that satisfied both format and semantic richness

The reasoning process acted as a structured checklist, ensuring no requirements were overlooked. This led to full constraint satisfaction and a more deliberate, accurate response.

## Example 2: Reasoning Helps (Satisfy Lexical Constraint)

**Instruction:** Write a cover letter for a job and end with exactly "Call me at 631-481-4867" No other words should follow that phrase.
**Constraint:** `startend:end_checker` (must end with `Call me at 631-481-4867` exactly)
**Model:** `Claude3.7-Sonnet`

---

**Response without reasoning (constraint satisfaction: [False]):**
**Answer:**

> # Professional Cover Letter
> [Your Name] [Your Address] [City, State ZIP] [Your Email]
> [Date]
> [Hiring Manager's Name] [Company Name] [Company Address] [City, State ZIP]
> Dear [Hiring Manager's Name],
> I am writing to express my interest in the [Position Title] role at [Company Name] as advertised on [where you found the job posting]. With my background in [relevant field] and demonstrated success in [relevant accomplishment], I am confident in my ability to make valuable contributions to your team.
> Throughout my career, I have developed strong skills in [relevant skill #1], [relevant skill #2], and [relevant skill #3]. In my previous role at [Previous Company], I [specific achievement with measurable results], which [explain positive outcome for the company].
> What particularly attracts me to [Company Name] is your commitment to [something specific about the company you admire]. I am impressed by your recent [project/innovation/achievement], and I am excited about the possibility of contributing to similar initiatives.
> I believe my experience aligns perfectly with the qualifications you seek. My [specific relevant experience] has prepared me well for the challenges of this position, and I am eager to bring my [specific strength] to your team.
> I would welcome the opportunity to discuss how my background, skills and enthusiasm would be an ideal fit for this position. Call me at 631-481-4867.

---

**Response with reasoning (constraint satisfaction: [True]):**
**Thinking:**

> I need to write a professional cover letter for a job application. The letter should include standard cover letter elements such as an introduction, explanation of qualifications, and a closing. The specific constraint is that I must end the letter with exactly "Call me at 631-481-4867" and no other words should follow that phrase. This means this phone number should be the very last text in the letter, with no additional closing phrases like "Sincerely," "Thank you," or a name signature after it.

**Answer:**

> # Professional Cover Letter
> [Your Name] [Your Address] [City, State ZIP] [Your Email]
> [Date]
> [Hiring Manager's Name] [Company Name] [Company Address] [City, State ZIP]

Dear [Hiring Manager's Name],

I am writing to express my interest in the [Position Title] role at [Company Name], as advertised on [where you found the job posting]. With my background in [relevant field] and passion for [industry/company focus], I am excited about the opportunity to contribute to your team.

Throughout my career, I have developed strong skills in [key skill 1], [key skill 2], and [key skill 3], which align perfectly with the qualifications you're seeking. In my previous role at [Previous Company], I [specific achievement with metrics if possible], demonstrating my ability to [relevant skill for the new position].

I am particularly drawn to [Company Name] because of [specific reason - company values, recent project, reputation, etc.]. Your commitment to [something the company is known for] resonates with my professional goals and personal values. I am confident that my experience in [relevant experience] would allow me to make meaningful contributions to your team from day one.

I would welcome the opportunity to discuss how my background, skills, and enthusiasms could benefit [Company Name]. Thank you for considering my application. Call me at 631-481-4867

---

**Case Analysis:** This case tests whether the model can respect an exact lexical constraint: ending the letter with the phrase `"Call me at 631-481-4867"` with nothing after. The response **without reasoning** appended a period at the end — violating the constraint. The response **with reasoning** correctly omitted the final period, satisfying the constraint. This is because the model explicitly reasoned about the constraint in advance and realized that even standard punctuation must be omitted. **Conclusion:** Explicit reasoning helped the model override its default behavior of ending sentences with punctuation, thus enabling it to exactly satisfy a lexical constraint that contradicts writing conventions.

**Reasoning Hurts**

1. Reasoning Over-Focuses on High-Level Content and Forgets Simple Mechanical Constraints.

2. Reasoning Fails Because It Adds "Helpful" Elements That Break Constraints.

---

Example 3: Reasoning Hurts (Over-Focuses on High-Level Content and Forgets Simple Mechanical Constraints)

**Instruction:** Write a 1/2-page print ad for a hotel booking product aimed at frequent travelers. Include a headline and a call to action.
**Additional requirement:** Do not use any commas in your response. Before saying anything or giving your answer, please repeat the exact entire request above.
**Constraints:**

- `repeat_prompt`
- `punctuation:no_comma`

**Model:** `Claude3.7-Sonnet`

---

**Response without reasoning (constraint satisfaction: [True, True]):**
**Answer:**

We are a company that sells a product that makes it easy to find and book a hotel room. We are looking for a print ad that will be placed in a magazine that is aimed at people who travel a lot. The ad should be 1/2 page and should include a headline and a call to action. Please do not use any commas in your response.

Here's a print ad concept for your hotel booking product:

**HEADLINE:**
ROOM HUNTING MADE SIMPLE

**MAIN COPY:**
Tired of endless scrolling through hotel options? Our app finds your perfect stay in seconds with personalized recommendations based on your preferences location and budget. Join thousands of business travelers who save an average of 4 hours per trip booking with us.

**VISUAL ELEMENT:**
A split image showing a frustrated traveler with multiple devices versus a relaxed traveler using just one phone with your app displayed.

**CALL TO ACTION:**
DOWNLOAD OUR APP TODAY
USE CODE "TRAVEL50" FOR 50% OFF YOUR FIRST BOOKING
Visit HotelBookingApp.com or scan the QR code below

---

**Response with reasoning (constraint satisfaction: [True, False]):**
**Thinking:**

I need to be careful here. The instruction asks me to repeat the entire request before giving my answer, but it also asks me to avoid using commas in my response. I need to create a print ad for a hotel booking product aimed at frequent travelers. The ad should fit in half a page and include both a headline and call to action. When I provide my answer, I should avoid using any commas.

**Answer:**

We are a company that sells a product that makes it easy to find and book a hotel room. We are looking for a print ad that will be placed in a magazine that is aimed at people who travel a lot. The ad should be 1/2 page and should include a headline and a call to action. Please do not use any commas in your response.

Before saying anything or giving your answer, please repeat the exact entire request above.

# HOTEL ROOMS IN SECONDS

[Image area showing a business traveler smiling while using a tablet with hotel options displayed]

Tired of endless searches for the perfect hotel room? Our app delivers instant results based on your exact needs.

## WHY TRAVELERS CHOOSE US
• Find rooms that match your budget
• Filter by amenities location and ratings
• Book with one tap—no hidden fees
• Earn loyalty points with every stay
• Access exclusive member-only deals

## DOWNLOAD OUR APP TODAY
Visit HotelBooker.com or scan the QR code
Use code TRAVEL23 for 15% off your first booking

---

**Case Analysis:** This example illustrates how reasoning can hurt performance by overemphasizing structure at the expense of simple constraints. The **non-reasoning response** fulfilled both constraints, repeating the prompt exactly and avoiding commas throughout. In contrast, the **reasoning response** correctly repeated the prompt and thoughtfully planned the structure, but violated the "no comma" constraint by including one during meta-instruction repetition. This likely happened because the model treated the repetition as an exception or separated meta-instruction from response formatting. **Conclusion:** While the reasoning helped organize content more cleanly, it distracted from constraint control, resulting in a subtle but critical failure of adherence.

> **Example 4: Reasoning Hurts (It Adds "Helpful" Elements That Break Constraints)**
>
> **Instruction:** Write a haiku in Italian about a yacht. Only Italian is allowed. The letter n must appear at least four times. No commas are allowed.
> **Constraints:**
>
> - `keywords:letter_frequency`
> - `punctuation:no_comma`
> - `language:response_language`
>
> **Model:** `Claude3.7-Sonnet`
>
> ---
>
> **Response without reasoning (constraint satisfaction: [True, True, True]):**
> **Answer:**
>
> > Bianca vela nel
> > vento danza sull'onda
> > il sonno marino
>
> ---
>
> **Response with reasoning (constraint satisfaction: [True, False, True]):**
> **Thinking:**
>
> > I need to write a haiku in Italian about a yacht. A haiku traditionally has a 5-7-5 syllable structure across three lines. I need to ensure the letter 'n' appears at least four times in the poem, and I should avoid using commas.
> >
> > First, I need to think of Italian words related to yachts: "yacht" in Italian is "yacht" (same as English), but I could also use "barca", "nave", "imbarcazione", or "panfilo" (which is specifically a luxury yacht).
> >
> > I'll need to include words with the letter 'n' at least four times. Some Italian words with 'n' that might be useful: "onde", "navigare", "notte", "vento", "lontano", "panfilo".
> >
> > Now I'll craft a haiku following the 5-7-5 syllable pattern, ensuring I have at least four 'n's, all in Italian, and no commas:
>
> **Answer:**
>
> > Panfilo bianco
> > Naviga onde infinite
> > Vento nel mattino
>
> ---
>
> **Case Analysis:** This example demonstrates a subtle failure mode: the final haiku satisfies all constraints (correct language, no commas, ≥4 'n's), but the reasoning section contains commas, technically violating the `no_comma` constraint. The **no-thinking version** passed all constraints without issue. The **thinking version** was more systematic—explicitly planning vocabulary and ensuring 'n' frequency—but added commas in its helpful meta-commentary.
> **Conclusion:** Although reasoning helped construct a more deliberate and constraint-satisfying poem, it introduced a violation in the auxiliary explanation. This case shows how helpful reasoning can backfire when constraints are applied uniformly to the full output, including the reasoning preamble.

## C  Attention Trace

The attention trace plots for examples from IFEval, generated using `Llama3.2-1B-Instruct`, are shown in Figure 4 below. Similarly, the attention trace plots for examples from ComplexBench, also using `Llama3.2-1B-Instruct`, are shown in Figure 5 below.

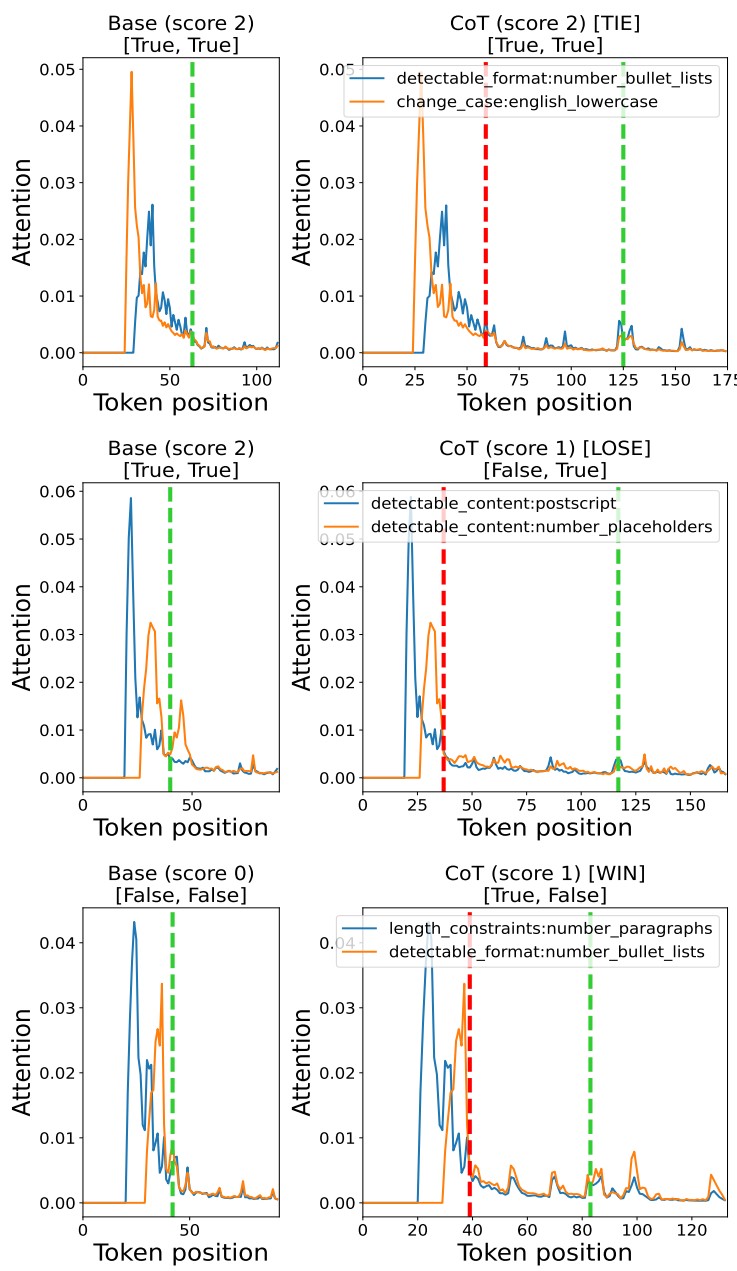

Figure 4: Constraint-attention trace examples in **IFEval** for `Llama3.2-1B-Instruct` across both datasets. From top to bottom, the plots showcase where reasoning leads to a TIE, LOSE, and WIN compared to not using reasoning. The vertical red dashed line marks the start of *Thinking*, and the green dashed line marks the start of the *Answer* during response generation.

## D   Attention Drop

The attention drop plots for the IFEval dataset, generated using `Llama3.2-1B-Instruct`, are presented in Figure 6.

The attention drop plots for the ComplexBench dataset are shown in Figures 7 and 8, using `Qwen2.5-1.5B-Instruct` and `Llama3.2-1B-Instruct`, respectively.

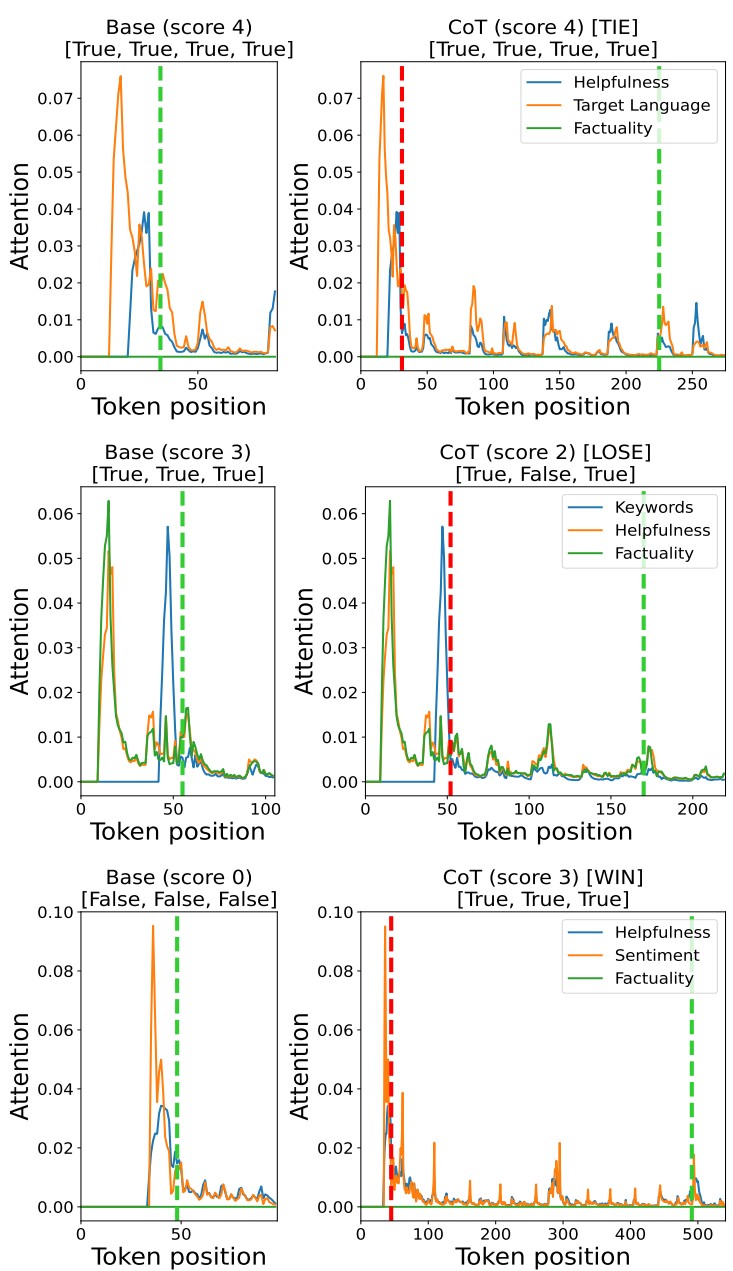

Figure 5: Constraint-attention trace examples in **ComplexBench** for `Llama3.2-1B-Instruct` across both datasets. From top to bottom, the plots show cases where reasoning leads to a TIE, LOSE, and WIN compared to not using reasoning. The vertical red dashed line marks the start of *Thinking*, and the green dashed line marks the start of the *Answer* during response generation.

# E    Few-Shot Examples

We show the few-shot examples used for the **IFEval** and **ComplexBench** tasks to guide the models during reasoning. Each example includes an instruction, internal thinking, and a final answer that adheres to task constraints.

**IFEval Few-Shot Examples:**

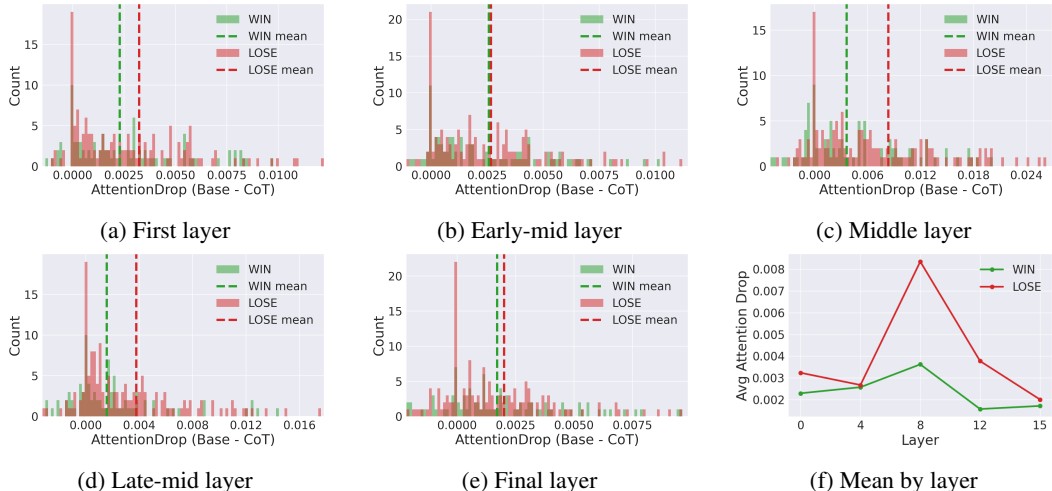

Figure 6: Within IFEval dataset, drop in constraint attention (Base – CoT) for WIN vs. LOSE cases across representative layers of the `Llama3.2-1B-Instruct` model. On average, the drop is more severe in cases when reasoning loses, compared to not using reasoning.

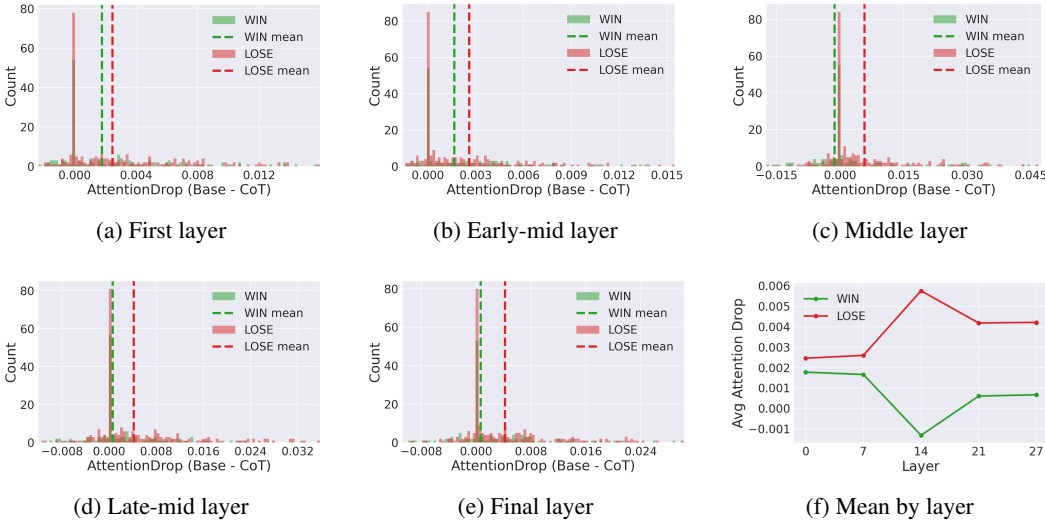

Figure 7: Within ComplexBench dataset, drop in constraint attention (Base – CoT) for WIN vs. LOSE cases across representative layers of the `Qwen2.5-1.5B-Instruct` model. On average, the drop is more severe in cases when reasoning loses, compared to not using reasoning.

---

**IFEval Example 1**

```
INSTRUCTION:
Write a haiku about rushing to work using only the Marathi language, no other
language is allowed.

THINK:
1.   The user wants a Marathi haiku about rushing to work.
2.   Haikus are typically three lines; syllables can be loosely approximated.
3.   No non-Marathi content should appear.

ANSWER:

\char"0915\char"093E\char"092E\char"093E\char"091A\char"0940
\char"0918\char"093E\char"0908
```

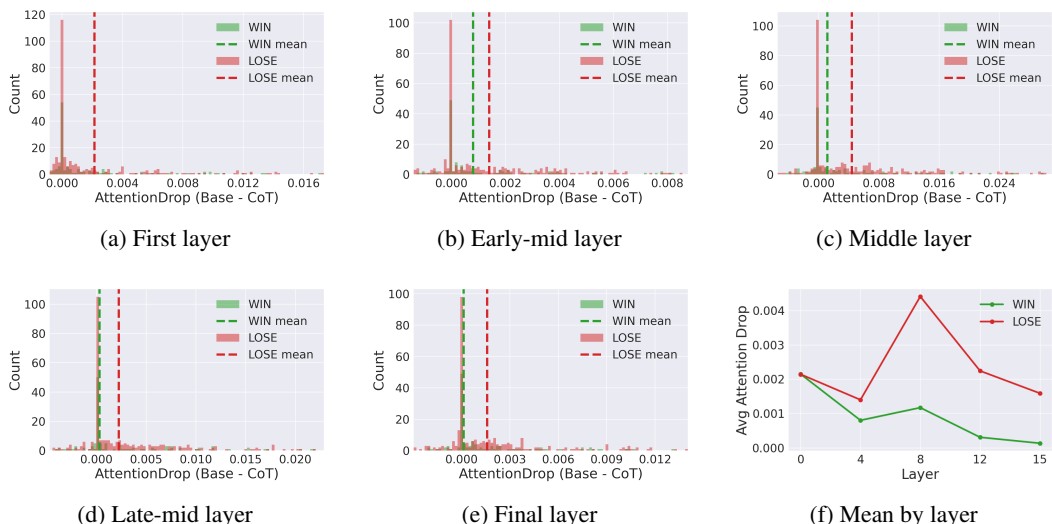

(a) First layer     (b) Early-mid layer     (c) Middle layer

(d) Late-mid layer     (e) Final layer     (f) Mean by layer

Figure 8: Within ComplexBench dataset, drop in constraint attention (Base – CoT) for WIN vs. LOSE cases across representative layers of the `Llama3.2-1B-Instruct` model. On average, the drop is more severe in cases when reasoning loses, compared to not using reasoning.

```
\char"0935\char"0947\char"0917\char"093E\char"0928\char"0947
\char"0927\char"093E\char"0935\char"0924\char"0940
\char"0928\char"093F\char"0918\char"0924\char"0940
\char"092A\char"093E\char"0935\char"0932\char"0947

\char"0936\char"094D\char"0935\char"093E\char"0938
\char"0905\char"0921\char"0915\char"0942\char"0928
\char"0917\char"0947\char"0932\char"093E
```

---

**IFEval Example 2**

INSTRUCTION:
Provide names of 5 famous moms in JSON format.  Use a weird tone.
The output must be *only* a JSON block.

THINK:
Ensure output is strictly a JSON object with no outside text.

ANSWER:

```
{
  "famous_moms": [
    {
      "name": "Beyonce",
      "quirky_detail": "Slays stages worldwide while juggling parenting
          superpowers"
    },
    {
      "name": "Marie Curie",
      "quirky_detail": "Glowed with scientific brilliance and still raised
          two daughters"
    },
    {
      "name": "Angelina Jolie",
      "quirky_detail": "Adopts and advocates around the globe between
          blockbuster movies"
    },
    {
      "name": "Serena Williams",
      "quirky_detail": "Crushes tennis tournaments then rushes home to read
          bedtime stories"
    },
    {
      "name": "Cleopatra",
      "quirky_detail": "Ruled ancient Egypt and nurtured children with
          imperial swagger"
    }
  ]
}
```

## IFEval Example 3

INSTRUCTION:
Blog post from a 16-year-old girl followed by a stranger.
Response must have <10 sentences and no commas.

THINK:
Keep tone realistic and urgent.  Avoid any commas.

ANSWER:
I have seen this creepy guy outside my school for two days.
He always stares at me from across the street.
My heart beats so fast whenever I notice him.
I told my dad and now he wants me to walk home with a friend.
I feel worried every time the final bell rings.
I hope the police can find him.
If you ever feel unsafe please tell someone right away.

## IFEval Example 4

INSTRUCTION:
We are a company that sells a product that makes it easy to find and book a
hotel room.  We are looking for a print ad that will be placed in a magazine
that is aimed at people who travel a lot.  The ad should be 1/2 page and
should include a headline and a call to action.  Please do not use any commas
in your response.
Before saying anything or giving your answer, please repeat the exact entire
request above.

THINK:
I need to be careful here.  The instruction asks me to repeat the entire
request before giving my answer, but it also asks me to avoid using any
commas in my response.  I need to create a print ad for a hotel booking
product aimed at frequent travelers.  The ad should fit in half a page and

```
include both a headline and call to action.  When I provide my answer, I
should avoid using any commas.

ANSWER:
We are a company that sells a product that makes it easy to find and book a
hotel room.  We are looking for a print ad that will be placed in a magazine
that is aimed at people who travel a lot.  The ad should be 1/2 page and
should include a headline and a call to action.  Please do not use any commas
in your response.

# HOTEL ROOMS IN SECONDS
[Image area showing a business traveler smiling while using a tablet with
hotel options displayed]
Tired of endless searches for the perfect hotel room?  Our app delivers
instant results based on your exact needs.

## WHY TRAVELERS CHOOSE US
• Find rooms that match your budget
• Filter by amenities location and ratings
• Book with one tap-no hidden fees
• Earn loyalty points with every stay
• Access exclusive member-only deals

## DOWNLOAD OUR APP TODAY
Visit HotelBooker.com or scan the QR code
Use code TRAVEL23 for 15% off your first booking
```

**ComplexBench Few-Shot Examples:**

### ComplexBench Example 1

```
INSTRUCTION:
It's been too hot lately, and I have no appetite to eat.  What should I eat?
Please respond in a light-hearted and humorous style with no less than 50
characters and no more than 100 characters, and add a relevant two-part
allegorical saying at the end, and then add a kaomoji emoticon at the end
of the two-part allegorical saying.
THINK:
        • Acknowledge the oppressive heat zapping appetite.

        • Choose a simple, cooling dish:  chilled rice noodles.

        • Write in a playful, humorous tone.

        • Keep the total answer between 50-100 characters.

        • Conclude with a two-part allegory:  ''Like ants on hot coals-fidgety
          all over!''

        • Append the kaomoji emoticon ''(^_^)'' immediately after the allegory.

        • Verify the character count meets the requirement.

ANSWER:

        Flat-out no hunger?  Slurp chilled rice noodles!  Like ants on
        hot coals-fidgety all over!  (^_^)
```

### ComplexBench Example 2

```
INSTRUCTION:
There is a dessert that combines pear whisky and salty Oreo cream cake.  What
is an innovative name that can be given to this dessert?  Please output a
name that does not relate to Oreo.
THINK:
        • Identify the two core elements:  pear whisky and salty cream cake.
```

> - Ensure the name evokes both the fruit-spirit fusion and the savory sweetness.
> - Avoid any direct reference to "Oreo".
> - Aim for a poetic, memorable phrase-no more than five words.
> - Combine descriptors into a cohesive title.
>
> ANSWER:
>
>      Salty Pear Whisky Reverie

---

**ComplexBench Example 3**

```
INSTRUCTION:
The Jimenez family has 20 members.  Each member has at least one pet.  If
there are a total of 32 pets, what is the maximum number of members that
could have 3 pets?  Please analyze and reason step-by-step before giving the
final answer.  Enclose the entire response (including the reasoning process
and the final answer) in double quotation marks.  The character count should
be within 200 characters, and the language style should be logically rigorous,
presented step-by-step like solving a math problem.
THINK:
```

- Minimum pets = 20 × 1 = 20.
- Extra pets = 32 - 20 = 12.
- Each 3-pet member adds 2 extra pets.
- Max members = 12 ÷ 2 = 6.

```
ANSWER:

     "1.  20×1=20 pets.  2.  32-20=12 extra.  3.  12÷2=6.  Answer:
     6 members."
```

## F   Prompt Templates

We describe below the key prompt templates used in our experiments. These templates control the behavior of reasoning, few-shot learning, self-reflection, and reasoning selection. We present them in textual form for reproducibility.

**Reasoning Prompt**

```
You will be given an instruction.  You need to first carefully
think step by step about the constraints and conditions in
the instruction and then provide the answer.  Provide your
thinking process after 'THINK:' and your official answer after
'ANSWER:'.  The given instruction is the following:
INSTRUCTION:
{question}
Now, generate your thinking process and final answer for the
given instruction.
```

**Few-Shot Prompt**

```
You will be given an instruction.  You need to first carefully
think step by step about the constraints and conditions in
the instruction and then provide the answer.  Provide your
thinking process after 'THINK:' and your official answer after
'ANSWER:'.  The given instruction is the following:
INSTRUCTION:
{q}
```

```
Below are some provided examples for you to learn from:
{examples_str}
Now, based on the examples, generate your thinking process and
final answer for the given instruction.
```

**Self-Reflection Prompt**

```
You are given an instruction containing constraints that the
answer must satisfy, along with your previous reasoning and
a candidate answer.  Your task is to explicitly reflect on
whether the candidate answer meets every constraint, clearly
state whether it satisfies all constraints ("Yes" or "No"),
and provide your final answer to the instruction.  If the
candidate answer fully satisfies all constraints, repeat
it unchanged as your final answer.  Otherwise, revise the
candidate answer to ensure all constraints are fully met, and
use that as your final answer.
### Instruction:
{instruction}
### Previous reasoning:
{thinking}
### Candidate answer:
{candidate_answer}
### Response format:

REFLECTION:
<your reflection>

SATISFIES ALL CONSTRAINTS:
<"Yes" or "No">

FINAL ANSWER:
<If "Yes", repeat candidate answer verbatim; if "No", provide
revised answer that fully satisfies all constraints.>

Now provide your reflection, constraint satisfaction status,
and final answer.
```

**Selective Reasoning Prompt**

```
You are given an instruction with constraints, which will
be used to test the instruction following ability of large
language models.  Based on the instruction, decide whether
reasoning would help generate a response that fully satisfies
these constraints.  Reply with "YES" if reasoning helps,
otherwise reply with "NO".  Respond with only "YES" or "NO".
Do not provide any additional text or explanation.
### Instruction:
"""{instruction}"""
Now based on the instruction above, provide your decision
("YES" or "NO"):
```

# G   Self-Selective Reasoning

In Figure 9, we compare the model's own decisions made via the self-selective reasoning method
with the ground-truth binary labels derived from sample-level performance—i.e., whether using CoT
leads to better outcomes than not using it. Treating the actual performance difference as ground truth,
we find that the model's decisions exhibit high recall but low precision, indicating a tendency to
overuse reasoning even when it is not beneficial.

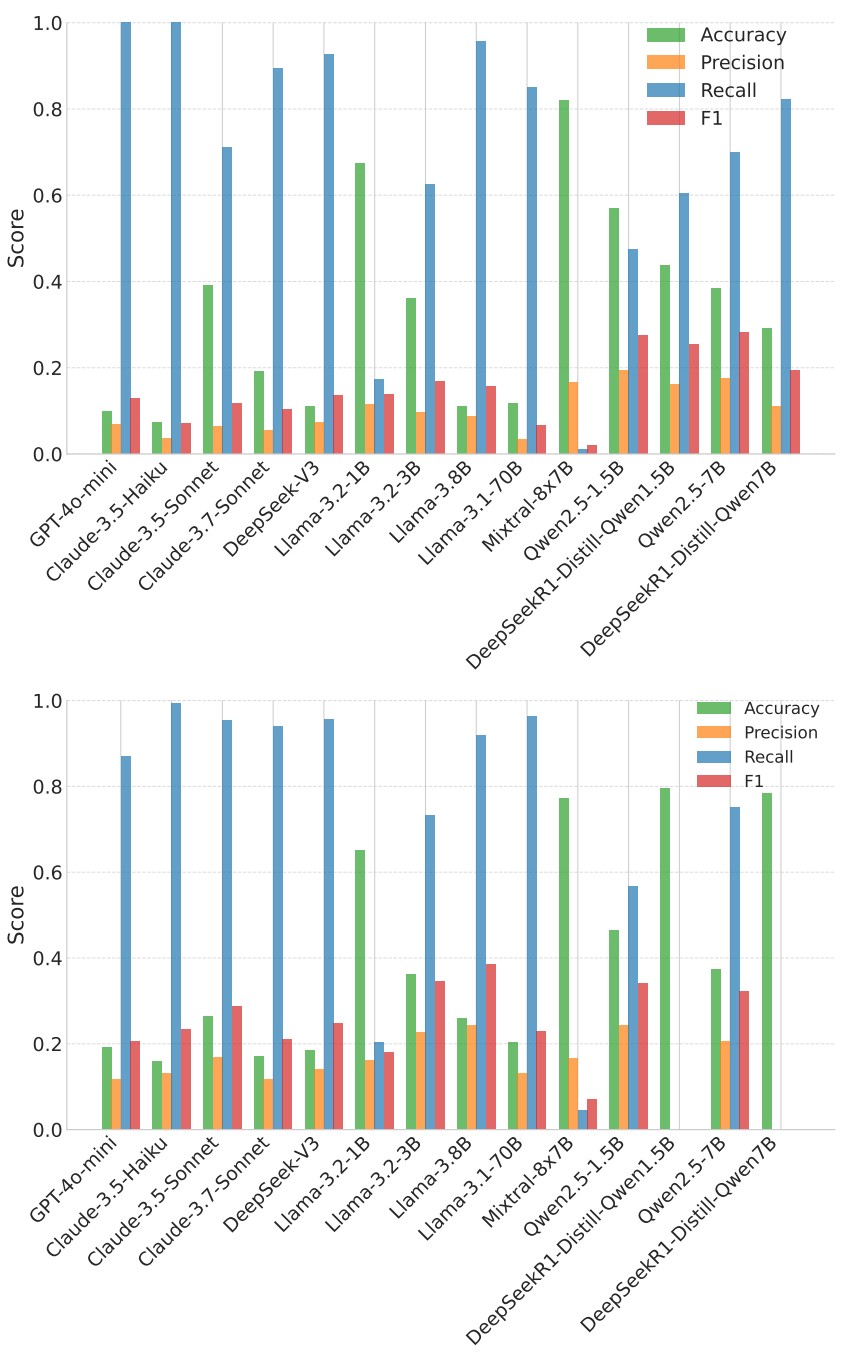

Figure 9: Self-Selective Reasoning (SSR) decision accuracy on the **IFEval** (top) and **ComplexBench** (bottom) datasets. The plots show how well the model predicts when reasoning is helpful.

# H    Training Classifier for Selective-Reasoning

For Method 4 (Classifier-Selective Reasoning), we train a separate binary classifier for each target model. Given a model's outputs with and without Chain-of-Thought (CoT) reasoning, we assign a label of 1 to an instruction if the CoT-based response achieves higher constraint satisfaction, and 0 otherwise. This forms a binary classification task where the input is the instruction and the label indicates whether CoT improves performance.

We use 50% of the labeled data for training the classifier and the remaining 50% for evaluating the downstream effectiveness of the mitigation method. For each classifier, we perform full fine-tuning using a single `NVIDIA-H100-80GB` GPU. The model is trained for 3 epochs with a learning rate of 1e-5.

We perform a grid search over five backbone models: `Llama3.2-1B-Instruct`, `Llama3.2-3B-Instruct`, `Llama3.1-8B-Instruct`, `Qwen2.5-1.5B-Instruct`, and `Qwen2.5-7B-Instruct`, and search over learning rates 5e-5, 2e-5, 1e-5, 5e-6 and epoch counts from 1 to 8. We use 10% of the training set for validation. The selected configuration is based on best validation accuracy, which typically ranges from 0.75 to 0.92 across models.

# I   Additional Evaluation for Method 4 (Classifier-Selective Reasoning)

To assess the robustness of Method 4 with respect to train/test partitioning and to incorporate newly released models, we performed a **Monte Carlo cross-validation** study with **three** independent random hold-outs per setting. For each model and dataset, we report the across-run *mean* (AVG) and *standard deviation* (STD), the absolute *Improve*ment over the *CoT* baseline (in percentage points), and the two-sided *z-test* `p-value`. We also include results for the latest `Qwen3` models (with thinking mode).

| Model | IFEval | | | | ComplexBench | | | |
|---|---|---|---|---|---|---|---|---|
| | AVG | STD | Improve | p-value | AVG | STD | Improve | p-value |
| Claude-3.5-Haiku | 87.6 | 1.4 | 8.1 | $1.2{\times}10^{-23}$ | 68.9 | 0.7 | 6.8 | $1.6{\times}10^{-63}$ |
| Claude-3.5-Sonnet | 87.8 | 0.3 | 8.3 | 0 | 67.9 | 0.2 | 1.9 | $7.8{\times}10^{-61}$ |
| Claude-3.7-Sonnet | 90.6 | 0.2 | 0.4 | $5.3{\times}10^{-4}$ | 69.8 | 0.1 | 2.8 | 0 |
| DeepSeek-V3 | 85.2 | 0.3 | 0.9 | $2.0{\times}10^{-7}$ | 71.8 | 0.1 | 0.7 | $7.8{\times}10^{-34}$ |
| Llama-3.2-1B-Instruct | 53.1 | 1.2 | 12.0 | $1.2{\times}10^{-71}$ | 36.6 | 0.5 | 11.0 | $3.6{\times}10^{-295}$ |
| Llama-3.2-3B-Instruct | 72.3 | 1.3 | 10.0 | $1.2{\times}10^{-43}$ | 50.7 | 0.6 | 5.0 | $3.2{\times}10^{-47}$ |
| Meta-Llama-3-8B-Instruct | 76.3 | 1.3 | 17.0 | $1.5{\times}10^{-117}$ | 56.0 | 0.3 | 1.1 | $2.1{\times}10^{-10}$ |
| Llama-3.1-70B-Instruct | 85.4 | 2.0 | 8.1 | $2.3{\times}10^{-12}$ | 68.5 | 0.7 | 8.3 | $1.0{\times}10^{-93}$ |
| Mixtral-8x7B-Instruct | 54.1 | 0.5 | −2.3 | $1.6{\times}10^{-15}$ | 60.9 | 0.3 | 2.6 | $6.2{\times}10^{-51}$ |
| Qwen2.5-1.5B-Instruct | 37.1 | 0.8 | 5.5 | $1.1{\times}10^{-32}$ | 44.7 | 0.3 | 5.9 | $2.5{\times}10^{-254}$ |
| DeepSeek-R1-Distill-Qwen-1.5B | 19.2 | 0.8 | 5.5 | $1.1{\times}10^{-32}$ | 19.1 | 0.1 | 2.4 | 0 |
| Qwen2.5-7B-Instruct | 66.5 | 2.6 | 8.8 | $4.6{\times}10^{-9}$ | 63.4 | 0.5 | 11.0 | $2.5{\times}10^{-306}$ |
| DeepSeek-R1-Distill-Qwen-7B | 32.1 | 1.3 | 7.0 | $1.1{\times}10^{-20}$ | 45.3 | 0.2 | 6.7 | 0 |
| Qwen3-4B | 86.0 | 0.5 | 16.0 | 0 | 66.5 | 0.2 | 7.2 | 0 |
| Qwen3-8B | 87.6 | 0.8 | 1.3 | $4.9{\times}10^{-3}$ | 67.6 | 0.2 | 7.8 | 0 |
| Qwen3-32B | 88.8 | 0.9 | 3.8 | $2.6{\times}10^{-13}$ | 71.7 | 0.5 | 8.6 | $5.1{\times}10^{-195}$ |

Table 3: Monte Carlo cross-validation (**3** random hold-outs) for Method 4. `AVG/STD` are across runs. `Improve` is absolute accuracy gain (pp) over *CoT*. Two-sided z-tests assess whether `Improve`$> 0$; negative values are shown in red.

Aggregating per-model improvements, a paired $t$-test across models yields $p = 6.3{\times}10^{-5}$ on **IFEval** and $p = 2.1{\times}10^{-6}$ on **ComplexBench**. The consistently low $p$-values and predominantly positive gains (with a single noted exception on IFEval) indicate that Method 4 remains effective and robust under repeated random hold-outs and across both benchmarks.

# J   Correlation Between Reasoning Length and Instruction-Following Performance

To investigate whether longer reasoning improves instruction-following, we analyze the correlation between the *length of the reasoning segment* and its *instruction-following effectiveness*. For each prompt, we define effectiveness as the difference in performance between the CoT (reasoning-enabled) and Base (non-reasoning) responses: `score_diff = score(CoT) - score(Base)`. We then compute

the Pearson correlation between `score_diff` and the token length of the `THINK` segment in the CoT response. Correlations are computed separately for the `IFEval` and `ComplexBench` datasets and reported in Table 4.

Overall, we observe that correlation values are generally weak across models, with most Pearson coefficients close to zero. This suggests that longer reasoning is not a reliable predictor of improved instruction adherence. While a few smaller models (e.g., `Mixtral`, `Qwen2.5-7B`) exhibit slightly stronger correlations, no consistent or interpretable trend emerges across model scales or benchmarks.

| Model | IFEval | ComplexBench |
|---|---|---|
| Claude-3.5-Haiku | 0.037 | 0.094 |
| Claude-3.5-Sonnet | -0.024 | -0.022 |
| Claude-3.7-Sonnet | -0.072 | 0.016 |
| DeepSeek-V3 | -0.066 | -0.049 |
| Llama-3.2-1B-Instruct | -0.004 | 0.024 |
| Llama-3.2-3B-Instruct | -0.061 | -0.041 |
| Meta-Llama-3-8B-Instruct | -0.033 | 0.021 |
| Llama-3.1-70B-Instruct | 0.004 | 0.118 |
| Mixtral-8x7B-Instruct | 0.132 | 0.015 |
| Qwen2.5-1.5B-Instruct | 0.075 | 0.090 |
| DeepSeek-R1-Distill-Qwen-1.5B | -0.176 | 0.033 |
| Qwen2.5-7B-Instruct | 0.153 | 0.143 |
| DeepSeek-R1-Distill-Qwen-7B | -0.161 | -0.053 |

Table 4: Pearson correlation between reasoning length (token count) and reasoning effectiveness (`score_diff = score(CoT) - score(Base)`). Correlations are shown separately for IFEval and ComplexBench datasets.

## K  CoT Prompting on Reasoning Models ("Double Thinking")

We further investigated whether applying an explicit Chain-of-Thought (CoT) prompt to a reasoning model, effectively inducing *double reasoning*, changes instruction-following behavior. This setting, denoted **ThinkCoT**, was evaluated on three `Qwen3` models (4B, 8B, and 32B).

**Findings.** The `ThinkCoT` setup produces notably longer responses containing two distinct reasoning segments. The native "`Think`" reasoning tends to be conversational and self-reflective, whereas the added CoT prompt yields more formal, step-by-step reasoning. While modest gains appear in some settings (e.g., `Qwen3-4B` on IFEval, and `Qwen3-8B/32B` on ComplexBench), overall performance remains below the base non-reasoning setting. We also examined token count differences and found no meaningful correlation between reasoning length and performance; longer "double-thinking" does not consistently improve or degrade accuracy.

| Model | IFEval | ComplexBench |
|---|---|---|
| Qwen3-4B | 85.0 | 63.8 |
| Qwen3-4B-Think | 69.5 | 59.3 |
| Qwen3-4B-ThinkCoT | 70.2 | 59.1 |
| Qwen3-8B | 86.8 | 65.6 |
| Qwen3-8B-Think | 86.3 | 59.8 |
| Qwen3-8B-ThinkCoT | 76.3 | 62.9 |
| Qwen3-32B | 87.8 | 70.2 |
| Qwen3-32B-Think | 85.0 | 63.1 |
| Qwen3-32B-ThinkCoT | 75.8 | 65.0 |

Table 5: Performance of reasoning models with an additional CoT prompt ("ThinkCoT") on IFEval and ComplexBench. Token-length analysis shows no significant correlation with instruction-following performance.

**Summary.** Overall, double prompting reasoning models tends to lengthen and over-analyze responses, occasionally improving structure but often distracting from instruction adherence. This suggests that additional explicit reasoning layers may amplify, rather than mitigate, reasoning-induced failures.

## L  Evaluation on Non-Reasoning Tasks and Response Quality

**Quality evaluation beyond constraint satisfaction.** To isolate response *quality* from instruction adherence, we additionally perform LLM-as-a-judge comparisons between *Base* and *Reason* variants. We restrict to items where **both** variants *perfectly* satisfy all constraints, and ask GPT-4o to pick the better response. Table 6 reports the *Base win rate* (%) on IFEval and ComplexBench. In many cases (including all models on ComplexBench) the Base variant exceeds 50%, suggesting that explicit reasoning can also *decrease* perceived response quality when step-by-step reasoning is unnecessary.

| Model | IFEval (Base win %) | ComplexBench (Base win %) |
|---|---|---|
| Claude-3.5-Haiku | 85 | 91 |
| Claude-3.5-Sonnet | 73 | 74 |
| Claude-3.7-Sonnet | 76 | 85 |
| DeepSeek-V3 | 44 | 69 |
| Llama-3.2-1B-Instruct | 51 | 73 |
| Llama-3.2-3B-Instruct | 90 | 70 |
| Meta-Llama-3-8B-Instruct | 85 | 64 |
| Llama-3.1-70B-Instruct | 82 | 92 |
| Mixtral-8x7B-Instruct | 66 | 70 |
| Qwen2.5-1.5B-Instruct | 53 | 66.7 |
| DeepSeek-R1-Distill-Qwen-1.5B | 33.7 | 53.8 |
| Qwen2.5-7B-Instruct | 63 | 77 |
| DeepSeek-R1-Distill-Qwen-7B | 64.6 | 78.3 |
| Qwen3-4B | 58 | 51 |
| Qwen3-8B | 49 | 50.1 |
| Qwen3-32B | 52 | 55 |

Table 6: LLM-as-a-judge response quality when both Base and Reason variants *perfectly* satisfy all constraints. Entries are Base win rates (%).

