# OpenReview forum: "When Thinking Fails: The Pitfalls of Reasoning for Instruction-Following in LLMs"
_NeurIPS.cc/2025/Conference — NeurIPS 2025 spotlight_

### Official Review · Reviewer_TQ5w · 2025-06-24

**Clarity:** 3
**Significance:** 3
**Originality:** 3
**Rating:** 5
**Confidence:** 4

**Summary:**

This paper identifies an interesting phenomenon, language model optimized for reasoning will usually have a surprising degradation on instruction-following ability. The authors analyzed why such failures occur, places where reasoning helps and places where reasoning might hurt.

The authors did comprehensive experiments on a wide set of models including reasoning and non-reasoning models, with or without CoT, and show that there's a consistent trend of instruction-following performance degradation across almost all models. In addition, the authors proposed several methods to help mitigate such failures, and showed that a classifier-selection based method can help bring back the ability on instruction-following.

**Questions:**

See above.

- Classifier-selective method: Can the authors split the data according to different patterns, train the classifier on a set of those patterns, and test on another set? In practice we would hope such classifier can generalize to a wide set of unseen patterns in order to be useful.
- Another simple mitigation could be training the model directly for instruction-following. Have the authors considered this (e.g., RL with rewards as the instruction is being followed vs not) and how would it compare with the classifier based method?

**Ethical Concerns:**

["NO or VERY MINOR ethics concerns only"]

**Final Justification:**

I've read the authors' rebuttal and it addressed most of my concerns, specifically the ones related to the generalization on hold-out IF tasks. I would maintain my current positive rating.

**Limitations:**

Discussed in Section 6.

**Quality:**

3

**Strengths And Weaknesses:**

Strengths:
- The paper is overall well-written.
- The authors identified an important phenomenon: the model becomes worse at instruction-following when optimized for reasoning, which is surprising and remains to be under-explored in existing literature. The authors also did comprehensive analysis on specific cases showing why the model might exhibit such behavior.
- Experiments are performed over a wide set of models and on two instruction-following benchmarks. The authors have also proposed four different mitigation methods and showed they yield various gains on the IF benchmarks.

Weaknesses:
- I wonder how the classifier-selective method generalizes in practice. It does yield good performance on the two benchmarks, but could be due to the classifier is trained to identify similar patterns (since the patterns on the two IF benchmarks are fairly limited), and learn to use or not use CoT on those patterns. Can the authors split the data according to different patterns, train the classifier on a set of those patterns, and test on another set? In practice we would hope such classifier can generalize to a wide set of unseen patterns in order to be useful.
- Another simple mitigation could be training the model directly for instruction-following. Have the authors considered this (e.g., RL with rewards as the instruction is being followed vs not) and how would it compare with the classifier based method?

---

> ### Author Rebuttal · Authors · 2025-07-29
>
> **TLDR**: *We evaluated the generalization of the classifier-selective method by training on specific constraint subsets, showing improved generalization when focusing on constraints where reasoning typically fails. Additionally, we explored direct instruction-following fine-tuning (SFT), finding it effective but also potentially harmful to performance on unrelated tasks, highlighting the importance of weight-free mitigation methods.*
>
> Thank you for recognizing both the novelty of the phenomenon and the comprehensive nature of our analysis and experiments. We also appreciate your acknowledgment of our mitigation strategies and their effectiveness across diverse models and benchmarks. Meanwhile, we sincerely appreciate the comments and provide our responses below:
>
> # 1. Generalizability of Classifier-Selective Method:
> We understand the concern about the generalizability of the classifier-selective method and agree that it is an important aspect to explore. One challenge is that each sample often contains multiple overlapping constraints. Nonetheless, we try our best to implement the idea of training the classifier on a subset of constraint types and evaluating its generalization to other types. Specifically, the IFEval dataset annotates constraints with higher-level categories including:
> * *['detectable_content', 'startend', 'combination', 'keywords', 'change_case', 'punctuation', 'detectable_format', 'language', 'length_constraints']*.
>
> We selected five types such that the samples containing these categories account for roughly half of the dataset:
> * *['length_constraints', 'combination', 'startend', 'language', 'change_case'].*
>
> For each model, we trained the classifier only on this subset and evaluated its performance on the held-out half.
> Interestingly, we observed even stronger results than those from classifiers trained on a random 50% split: **in many cases, the improvements were significantly larger than before**, as shown below (improvements are relative to the Reason/CoT model, save as in the tables in paper):
>
> | Model                             | Score | Improvement |
> |-----------------------------------|-------|-------------|
> | GPT-4o-mini                       | 86.0  | 9.1         |
> | Claude-3.5-Haiku                  | 89.7  | 10.2        |
> | Claude-3.5-Sonnet                 | 88.5  | 9.0         |
> | Claude-3.7-Sonnet                 | 91.1  | 0.9         |
> | Deepseek-V3                       | 89.7  | 5.4         |
> | Llama-3.2-1B-Instruct             | 59.0  | 18.3        |
> | Llama-3.2-3B-Instruct             | 79.3  | 17.4        |
> | Meta-Llama-3-8B-Instruct          | 77.1  | 18.1        |
> | Llama-3.1-70B-Instruct            | 86.7  | 9.4         |
> | Mixtral-8x7B-Instruct             | 63.8  | 7.4         |
> | Qwen2.5-1.5B-Instruct             | 47.2  | 15.6        |
> | DeepSeek-R1-Distill-Qwen-1.5B     | 24.7  | 11.0        |
> | Qwen2.5-7B-Instruct               | 69.4  | 11.7        |
> | DeepSeek-R1-Distill-Qwen-7B       | 43.2  | 18.1        |
> | Qwen3-4B                          | 85.2  | 15.7        |
> | Qwen3-8B                          | 88.6  | 2.3         |
> | Qwen3-32B                         | 88.9  | 3.9         |
>
> First, these results show that the **classifier-selective method does generalize across constraint types** in certain aspects. Interestingly, they also suggest that **training the classifier specifically on certain constraint types may lead to even better overall performance**.
>
> To explain this, we hypothesize that it is because the selected types (e.g., *length_constraints*, *combination*) are where CoT is most prone to introducing errors. To support this, we **conducted a further quantitative analysis of constraint-type distributions**. Below we show the Reason-WIN vs. Reason-LOSE distributions on IFEval for two representative models (detailed distributions for all models/tasks will be included in the Appendix of our paper):
>
> **GPT-4o-mini**
>
> * *Reason WIN:* keywords 19.5%, length_constraints 17.1%, change_case 24.4%, combination 2.4%, detectable_format 19.5%, detectable_content 9.8%, punctuation 4.9%, startend 2.4%
>
> * *Reason LOSE:* combination 35.2%, length_constraints 31.0%, change_case 5.6%, keywords 16.9%, startend 2.8%, detectable_content 2.8%, punctuation 4.2%, detectable_format 1.4%
>
> **Claude-3.5-haiku**
>
> * *Reason WIN*: change_case 30.0%, keywords 25.0%, length_constraints 15.0%, detectable_format 20.0%, detectable_content 5.0%, combination 5.0%
>
> * *Reason LOSE*: combination 43.3%, length_constraints 25.4%, keywords 10.4%, startend 4.5%, change_case 4.5%, detectable_content 1.5%, detectable_format 6.0%, language 3.0%, punctuation 1.5%
>
> For both models, we observe a clear pattern: **Reason-WINs are concentrated in lexical or formatting categories** (e.g., *keywords*, *change_case*, *detectable_format*), while **Reason-LOSEs are dominated by global or compositional constraints** (e.g., *combination*, *length_constraints*, *startend*). **These “failure-prone” types were exactly those we selected for training the classifier, and moreover, these results match and further support our case study insights in Section 4.1**.
>
> # 2. Instruction-Following Training:
>
> We thank the reviewer for this thoughtful suggestion! Indeed, we are actively exploring additional methods to mitigate the drop in instruction-following performance caused by reasoning/CoT. The primary goal of this paper is to reveal and analyze this degradation, supported by quantitative experiments, case studies, and attention analysis. For mitigation, we focused on simple methods that do not alter model weights, because for methods involving training, which change the model weights, our major concern is how this training may affect performance on other tasks.
>
> We explore this with experiments. Certain benchmark datasets (such as ComplexBench) involve both rule-based evaluation and LLM-as-a-judge, so RL training for instruction-following might be expensive, as LLM-as-a-judge could significantly slow down the process (but we will explore this further in future work). Therefore, we **conducted supervised fine-tuning (SFT) experiments** using data labeled by the strongest model in our evaluation (Claude 3.7-Sonnet). We selected only samples where the model response received a perfect instruction-following score, and trained using these examples as positive supervision.
>
> We split half of the IFEval dataset for SFT training and evaluated instruction-following performance on the other half after SFT (denoted as “IFEval (SFT)”). We also evaluated performance before and after SFT on other representative tasks: ARC, TruthfulQA, GSM8K, and MinervaMath, which include both general knowledge and reasoning evaluation, to assess any unintended impact. Below are the results:
>
> | Model                               | Original | CoT   | IFEval(SFT) | ARC  | ARC(SFT) | TruthfulQA | TruthfulQA(SFT) | GSM8K | GSM8K(SFT) | MinervaMath | MinervaMath(SFT) |
> |-------------------------------------|----------|-------|---------------------|------|-----------|-------------|-------------------|--------|-------------|--------------|--------------------|
> | Llama-3.2-1B-Instruct               | 49.1    | 40.6 | 40.5                | 31.4 | 32        | 24.1        | 27.1              | 20.3   | 13.7        | 14.5         | 13.1               |
> | Meta-Llama-3-8B-Instruct           | 75.1    | 58.8 | 75                  | 52.8 | 46.7      | 40.5        | 34.6              | 80     | 75.2        | 29.6         | 28.4               |
> | Qwen2.5-1.5B-Instruct              | 35.7    | 31.7  | 38.3                | 38.7 | 36.9      | 31.2        | 30.9              | 63.3   | 63.2        | 16.8         | 18                 |
> | DeepSeek-R1-Distill-Qwen-1.5B      | 16.6    | 13.5 | 23.5                | 27.7 | 27.7      | 35.4        | 35.3              | 14.2   | 14          | 5.6          | 5                  |
> | DeepSeek-R1-Distill-Qwen-7B        | 30.6    | 25.0 | 36.4                | 32.7 | 32.5      | 30.7        | 29.4              | 9.9    | 12.9        | 6.3          | 8.5                |
>
> We observe that **SFT can significantly improve instruction-following performance** for many models (e.g., a +16.0 improvement for Meta-Llama-3-8B-Instruct). However, it **can also negatively impact performance on other tasks** (possibly due to catastrophic forgetting on the other tasks after SFT on instruction-following). For instance, Meta-Llama-3-8B-Instruct shows performance drops across all evaluated benchmarks after SFT. This highlights the need for caution when applying weight-updating mitigation strategies.
>
> Meanwhile, we are continuing to explore alternative mitigation strategies that preserve model weights. To share some ideas,  some directions include:
> * *Activation probing as a reasoning gate*: A lightweight linear probe trained on mid-layer activations could predict whether reasoning is needed. If reasoning is deemed unnecessary, the model uses the base response; otherwise, it switches to CoT.
> * *Logit-lens comparison*: By running Base and Reason side-by-side, we can use the logit lens to identify positions in the prompt where their token distributions diverge most. Examining these hidden-state differences could reveal the essential source of instruction-following failures.
> * *Activation steering to mitigate failures*: We could precompute steering vectors associated with constraint types (or derived from the hidden-state differences above) and inject them during generation. This may nudge the model toward better instruction adherence, without fine-tuning.
>
>
> Overall, we deeply appreciate the reviewer’s insightful comments and valuable suggestions. We sincerely hope our clarification and additional experiments above would address the concerns, and respectfully hope the reviewer could even further raise the score.

---

> > ### Comment · Reviewer_TQ5w · 2025-08-04
> >
> > Thanks for the rebuttal. I've read the authors' response and it addressed most of my concerns, specifically the ones related to the generalization on hold-out IF tasks. I would maintain my current positive rating.

---

> > > ### Author Response · Authors · 2025-08-04
> > >
> > > We sincerely thank the reviewer for the positive feedback! We are especially grateful for the insightful suggestions regarding the generalizability of the classifier-selective method and the consideration of model training for mitigation, both of which led to valuable additions to our work. We also truly appreciate the reviewer’s thoughtful evaluation of our paper and active engagement throughout the discussion process.

---

> > ### Comment · Reviewer_SzjC · 2025-08-04
> > **Response to Authors**
> >
> > Thanks to the authors for their elaborated response. I will keep my original score, suggesting to accept the paper to the conference.

---

### Official Review · Reviewer_EvS7 · 2025-06-30

**Clarity:** 3
**Significance:** 2
**Originality:** 2
**Rating:** 4
**Confidence:** 4

**Summary:**

The paper investigates a counterintuitive phenomenon where explicit Chain-of-Thought (CoT) reasoning can impair the ability of Large Language Models (LLMs) to follow instructions.

The paper uses two setups:
1) a model is prompted with and without a specialized prompt for the model to provide thinking before the final answer
2) a model is used with/without its native thinking mode (e.g. claude sonnet with and without thinking)

The authors show that performance tend to drop when thinking is used in two IF datasets.

My main concerns are that:
* setup (1) is not valid imo to assess the reasoning abilities of models, as models are prompted with a specialized instruction to think, which they most likely weren't trained to tackle. The point is that we shouldn't expect this specific prompt to unlock thinking in models.
* I think setup (2) is much more valid, as it compares a model with its native thinking behavior. My problem here is that results comparing thinking on/off are pretty close, and there is so significance test, so we cannot know whether there is any model degradation.

Given these weaknesses I think the paper requires more work

**Questions:**

* are the results for table 2 significant (i.e., is the performance when thinking is on significantly worse)?

**Ethical Concerns:**

["NO or VERY MINOR ethics concerns only"]

**Final Justification:**

The rebuttal addressed my concerns.

**Limitations:**

yes

**Quality:**

2

**Strengths And Weaknesses:**

Strengths:
* The central claim that reasoning can degrade instruction-following accuracy is unexpected and important.
* In-depth Analysis of Failure Modes: The paper does not just report the phenomenon but offers a deep dive into why it occurs. It provides a qualitative analysis through a large-scale manual case study, identifying recurring patterns where reasoning helps (e.g., formatting, lexical precision) and where it hurts (e.g., neglecting simple constraints, introducing unnecessary content).

Weaknesses (copying first 2 bullet points from above):
* setup (1) is not valid imo to assess the reasoning abilities of models, as models are prompted with a specialized instruction to think, which they most likely weren't trained to tackle. The point is that we shouldn't expect this specific prompt to unlock thinking in models.
* I think setup (2) is much more valid, as it compares a model with its native thinking behavior. My problem here is that results comparing thinking on/off are pretty close, and there is so significance test, so we cannot know whether there is any model degradation.
* Limited Scope of Investigation: The study's conclusion is confined to instruction-following evaluation (formatting and other simple constraints). What about addressing the actual quality of the responses?
* I think the results for methods to mitigate degradation (section 5) are not very informative as they are only applied to setup (1), which is problematic as I mentioned above.

---

> ### Author Rebuttal · Authors · 2025-07-29
>
> **TLDR:** *We clarify the validity of our two experimental settings, added new controlled experiments to reinforce our findings, conducted rigorous statistical significance tests to demonstrate robustness, and provided additional evaluations showing reasoning may not only reduce instruction-following accuracy but can also degrade overall response quality.*
>
> Thank you for highlighting the novelty and importance of our main finding. We also appreciate your recognition of the depth and rigor of our failure mode analysis and manual case study. Below are our responses to the comments:
>
> # 1. Design of Setting 1 and Setting 2
>
> First, we would like to clarify our motivation and thoughts behind the design of two experimental settings. **Setting 1 (CoT prompting)** involves using the same underlying model with and without chain-of-thought prompting. This provides a **well-controlled comparison since the model weights are identical, isolating the effect of prompting alone**. CoT prompting is also a common technique applied in practice to elicit reasoning. As noted in Section 3.4, **Setting 2 (comparing a base model to its Reasoning counterpart)** introduces potential confounds: Reason models may **undergo additional training stages on new datasets**, such as supervised fine-tuning or RL. For example, *DeepSeek-V3* and *DeepSeek-R1* differ significantly in training scale, with R1 having been further trained on much more data in multiple extra training stages, involving both SFT and RL. Despite these differences, we **include both settings to demonstrate the general phenomenon: a consistent drop in instruction-following performance after applying CoT or reasoning**.
>
> In general, we agree and acknowledge that each setting has distinct advantages and limitations. To further strengthen the Setting 2 results, we added additional base–reasoning pairs:
> * *Qwen2.5-1.5B-Instruct vs. Qwen2.5-Math-1.5B*
> * *Qwen2.5-7B-Instruct vs. Qwen2.5-Math-7B*
>
> In each case, the Math variant is explicitly fine-tuned for reasoning. Furthermore, we incorporated newly released Qwen3 models, which include both Base and Think modes. This is a better controlled experiment as the think vs. no-think mode only differ in the template and the model weights are the same:
>
> * *Qwen3-4B vs. Qwen3-4B-Think*
> * *Qwen3-8B vs. Qwen3-8B-Think*
> * *Qwen3-32B vs. Qwen3-32B-Think*
>
> We evaluate all pairs on both IFEval and ComplexBench. As shown in the table below, the **general trend persists: reasoning variants tend to underperform their base counterparts in instruction-following, across both benchmarks**.
> | Model                   | IFEval | ComplexBench |
> |-------------------------|--------|--------------|
> | Qwen3-4B                | 85.0   | 63.8         |
> | Qwen3-4B-Think          | 69.5   | 59.3         |
> | Qwen3-8B                | 86.8   | 65.6         |
> | Qwen3-8B-Think          | 86.3   | 59.8         |
> | Qwen3-32B               | 87.8   | 70.2         |
> | Qwen3-32B-Think         | 85.0   | 63.1         |
> | Qwen2.5-1.5B-Instruct   | 35.9   | 44.1         |
> | Qwen2.5-Math-1.5B       | 14.9   | 23.4         |
> | Qwen2.5-7B-Instruct     | 63.6   | 60.2         |
> | Qwen2.5-Math-7B         | 27.9   | 28.8         |
>
> We sincerely appreciate the reviewer’s comment and will incorporate these additional results and corresponding discussion into Section 3.4 of the revised paper.
>
> # 2. Significance Test
> In the evaluation setup, the rule-based Python functions are deterministic. For ComplexBench, which includes both rule-based and LLM-as-a-judge components, we also use temperature 0 for GPT-based evaluation. For the Classifier-Selective mitigation method, since it involves classifier training on a randomly sampled subset of the data, we further examine its robustness by running **Monte Carlo cross-validation by repeating random hold-out three times**. Below we report the **average**, **standard deviation**, **improvement** over the Reason/CoT model, and **p-values** from two-sided z-tests:
>
> | Model                         |      |      |  IFEval |            |      |      |ComplexBench   |            |
> |-------------------------------|------|------|---------|------------|------|------|---------|------------|
> |                               | AVG  | STD  | Improve | p-value    | AVG  | STD  | Improve | p-value    |
> | GPT-4o-mini                   | 83.4 | 1.4  | 6.5     | 8.9e-16    | 66.0 | 0.8  | 5.7     | 5.5e-35    |
> | Claude-3.5-Haiku              | 87.6 | 1.4  | 8.1     | 1.2e-23    | 68.9 | 0.7  | 6.8     | 1.6e-63    |
> | Claude-3.5-Sonnet             | 87.8 | 0.3  | 8.3     | 0          | 67.9 | 0.2  | 1.9     | 7.8e-61    |
> | Claude-3.7-Sonnet             | 90.6 | 0.2  | 0.4     | 0.00053    | 69.8 | 0.1  | 2.8     | 0          |
> | Deepseek-V3                   | 85.2 | 0.3  | 0.9     | 2e-07      | 71.8 | 0.1  | 0.7     | 7.8e-34    |
> | Llama-3.2-1B-Instruct         | 53.1 | 1.2  | 12      | 1.2e-71    | 36.6 | 0.5  | 11      | 3.6e-295   |
> | Llama-3.2-3B-Instruct         | 72.3 | 1.3  | 10      | 1.2e-43    | 50.7 | 0.6  | 5       | 3.2e-47    |
> | Meta-Llama-3-8B-Instruct      | 76.3 | 1.3  | 17      | 1.5e-117   | 56.0 | 0.3  | 1.1     | 2.1e-10    |
> | Llama-3.1-70B-Instruct        | 85.4 | 2.0  | 8.1     | 2.3e-12    | 68.5 | 0.7  | 8.3     | 1e-93      |
> | Mixtral-8x7B-Instruct         | 54.1 | 0.5  | -2.3    | 1.6e-15    | 60.9 | 0.3  | 2.6     | 6.2e-51    |
> | Qwen2.5-1.5B-Instruct         | 37.1 | 0.8  | 5.5     | 1.1e-32    | 44.7 | 0.3  | 5.9     | 2.5e-254   |
> | DeepSeek-R1-Distill-Qwen-1.5B | 19.2 | 0.8  | 5.5     | 1.1e-32    | 19.1 | 0.1  | 2.4     | 0          |
> | Qwen2.5-7B-Instruct           | 66.5 | 2.6  | 8.8     | 4.6e-09    | 63.4 | 0.5  | 11      | 2.5e-306   |
> | DeepSeek-R1-Distill-Qwen-7B   | 32.1 | 1.3  | 7       | 1.1e-20    | 45.3 | 0.2  | 6.7     | 0          |
> | Qwen3-4B                      | 86.0 | 0.5  | 16      | 0          | 66.5 | 0.2  | 7.2     | 0          |
> | Qwen3-8B                      | 87.6 | 0.8  | 1.3     | 0.0049     | 67.6 | 0.2  | 7.8     | 0          |
> | Qwen3-32B                     | 88.8 | 0.9  | 3.8     | 2.6e-13    | 71.7 | 0.5  | 8.6     | 5.1e-195   |
>
> We also computed the **overall p-values across all models using paired t-tests** on the improvement scores: **6.3e-05 for IFEval, and 2.1e-06 for ComplexBench**.
> These consistently small p-values, along with the positive improvement scores, indicate that the improvements from the Classifier-Selective method are statistically significant and robust. This further strengthens the effectiveness of our proposed approach.
>
> # 3.  Quality (besides Instruction Following) of Response:
> The overall response quality, beyond instruction-following performance, is indeed another important and interesting dimension to evaluate. We agree with the reviewer that reasoning can influence both instruction adherence and general response quality. While reasoning models often excel at tasks like math and code generation, the instructions in IFEval and ComplexBench are more about following details and less creative (e.g. writing an email in a specific format).
>
> In our paper, we focused on how reasoning can degrade instruction-following ability. Here, we **add new experiments to assess response quality** directly. Specifically, we use LLM-as-a-judge (GPT4o) to compare the responses of Base vs. Reason models. Importantly, we restrict this evaluation to only those examples where both models perfectly satisfy all constraints, so that differences in quality are not due to instruction-following violations.
>
> | Model                                 | IFEval | ComplexBench |
> |---------------------------------------|--------|---------------|
> | GPT-4o-mini                           | 87%  | 87%         |
> | Claude-3.5-Haiku                      | 85%  | 91%         |
> | Claude-3.5-Sonnet                     | 73%  | 74%         |
> | Claude-3.7-Sonnet                     | 76%  | 85%         |
> | Deepseek-V3                           | 44%  | 69%         |
> | Llama-3.2-1B-Instruct                 | 51%  | 73%         |
> | Llama-3.2-3B-Instruct                 | 90%  | 70%         |
> | Meta-Llama-3-8B-Instruct              | 85%  | 64%         |
> | Llama-3.1-70B-Instruct                | 82%  | 92%         |
> | Mixtral-8x7B-Instruct                 | 66%  | 70%         |
> | Qwen2.5-1.5B-Instruct                 | 53%  | 66.7%         |
> | DeepSeek-R1-Distill-Qwen-1.5B         | 33.7%  | 53.8%         |
> | Qwen2.5-7B-Instruct                   | 63%  | 77%         |
> | DeepSeek-R1-Distill-Qwen-7B           | 64.6%  | 78.3%         |
> | Qwen3-4B                              | 58%  | 51%         |
> | Qwen3-8B                              | 49%  | 50.1%         |
> | Qwen3-32B                             | 52%  | 55%         |
>
> We find that in many cases (even all models on ComplexBench) the base model achieves a win rate above 50%, indicating that **it produces higher-quality responses than the reasoning variant**, even when both follow instructions perfectly. This result suggests that **reasoning can not only hurt instruction following, but may also degrade general response quality** in tasks where step-by-step reasoning is unnecessary.
>
> Overall, we deeply appreciate the reviewer’s insightful comments and valuable suggestions. We sincerely hope our clarification and additional experiments above would address the concerns, and respectfully hope the reviewer could raise the score.

---

> ### Author Response · Authors · 2025-08-06
>
> Dear Reviewer EvS7,
>
> Thank you again for taking the time to review our paper. We hope that our rebuttal has addressed your concerns and clarified our contributions. If you find our response helpful in resolving the issues raised, we would be sincerely grateful if you would consider updating your score to reflect your current assessment. If any further questions or uncertainties remain, we would be more than happy to further clarify during the discussion period!

---

> ### Comment · Reviewer_EvS7 · 2025-08-07
>
> Thank you for addressing my concerns. I'm updating my score accordingly

---

> > ### Author Response · Authors · 2025-08-07
> >
> > We sincerely thank Reviewer EvS7 for the kind reply. We are very glad to hear that our rebuttal helped address your concerns, and we truly appreciate your consideration in updating the score.
> >
> > Moreover, we are especially grateful for your insightful comments on the experimental design, the significance of the results, and the general quality of the model’s responses. Your feedback has led to valuable improvements in our work. We deeply appreciate your thoughtful evaluation and engagement throughout the discussion process.

---

### Official Review · Reviewer_SzjC · 2025-07-01

**Clarity:** 3
**Significance:** 3
**Originality:** 3
**Rating:** 5
**Confidence:** 4

**Summary:**

The paper examines when chain of thought reasoning in LLMs leads to degrading performance. To this end, they show that explicit chain of thought reasoning can harm instruction following accuracy by systematically evaluating responses with and without chain of thought; applying the method on two benchmarks to a variety of LLMs from different LLM families from 1 to 70 B-parameters shows performance drops in most models. To mitigate the effect, the authors analyze individual reasoning scenarios and introduce an attention-based metric to quantify the influence in these scenarios, such as few-shot in-context learning and self-reflection.

**Questions:**

What other methods from mechanistic interpretability (MI) could be used to analyze attention traces or relevant parameters and the im MI so-called residual stream during generation?

How can similar issues arisong from (mis-) using chain of thought patterns be mitigated by combining different strategies, such as self-reflection or reasoning in probability space, or constrained reasoning?

**Ethical Concerns:**

["NO or VERY MINOR ethics concerns only"]

**Final Justification:**

I will keep my original score, suggesting to accept the paper to the conference.

**Limitations:**

Yes

**Quality:**

3

**Strengths And Weaknesses:**

The paper is well written and tackles a highly interesting topic: reasoning and language models, specifically where it might not be helpful.

While methodologically the paper just compares explicit chain of thought with no chain of thought, it carries out a thorough comparison and evaluation across a variety of language models and also reasoning scenarios.

It proposes an interesting approach to analyze attention patterns using attention interpretability methods, and shows illustrative visualizations of the relevant tokens, where the relevant tokens are identified using GPT-4. These relevant tokens are clustered into rules that are derived from the two benchmarks, giving the possibility for an interesting fine-grained analysis. (The statistics derived from this analysis include the attention trace and also layer-wise statistics, showing that chain of thought reasoning flattens the constraint-attention trace.)

Following their evaluation it derives concrete best practices for using reasoning in different scenarios, making the approach easily applicable.

Weaknesses:
My main concerns are with the generalizability of the approach:
First, the approach focuses only on English it would be interesting to see how it performs in other languages, maybe less resourced languages.
Second, I would find the approach more generalizable if it was also evaluated on tasks not specifically designed for reasoning.

There is (recent, but also older) related work that tackles similar aspects of reasoning, e.g., chain of though length, [1, 2, 3] that would be important to include and discuss here.

[1] https://aclanthology.org/2024.acl-long.818.pdf
[2] https://arxiv.org/pdf/2502.07266
[3] https://arxiv.org/pdf/2410.21333
[4] https://arxiv.org/pdf/2502.01100

---

> ### Author Rebuttal · Authors · 2025-07-29
>
> **TLDR**: *We agree exploring multilingual instruction-following evaluation is valuable future work. We’ve clarified that our benchmarks already include everyday tasks (not just reasoning-heavy scenarios), added quality-based evaluations using LLM judges, and incorporated correlation analysis regarding CoT length. Additionally, we discussed further mechanistic interpretability methods and acknowledged the potential of combining multiple mitigation strategies.*
>
> Thank you for the positive and detailed feedback. We’re especially glad that you found our evaluation thorough, the constraint-attention analysis insightful, and the methods practical and applicable. We also really appreciate the insightful comments! Below are our responses:
>
> # 1. Task on Non-English Languages
> We really appreciate this suggestion. We agree it would be interesting to study this phenomenon and apply our approaches to other languages. It is out of the scope of this paper but we will actively explore this direction and include multilingual analysis as part of our future work!
>
> # 2. Evaluation on Non-Reasoning Tasks
> If the reviewer means applying our attention-based analysis to non-reasoning tasks, we believe it can generalize. In our study, attention is used to identify peak signals on important tokens: specifically, tokens corresponding to constraints in instruction-following tasks. We examine how the model’s responses relate to these tokens through their attention scores. This approach is not specific to reasoning and should extend naturally to other tasks by redefining what constitutes “important” tokens.
>
> If the reviewer is referring to our instruction-following evaluations, in fact, here the IFEval and ComplexBench benchmarks are not specifically designed for reasoning; rather, they focus on evaluating instruction-following capabilities of LLMs. Most instructions in these datasets are everyday user requests (e.g., writing emails or formatting text), not reasoning-heavy tasks. If the reviewer means evaluating models on other aspects of performance beyond instruction following, we have also **added new experiments to assess response quality**. Specifically, we use **LLM-as-a-judge (GPT4o) to compare the responses of Base vs. Reason models**. We restrict this evaluation to only those examples where **both models perfectly satisfy all constraints**, so that differences in quality are not due to instruction-following violations. Below are the win rates of Base models:
> | Model                                 | IFEval | ComplexBench |
> |---------------------------------------|--------|---------------|
> | GPT-4o-mini                           | 87%  | 87%         |
> | Claude-3.5-Haiku                      | 85%  | 91%         |
> | Claude-3.5-Sonnet                     | 73%  | 74%         |
> | Claude-3.7-Sonnet                     | 76%  | 85%         |
> | Deepseek-V3                           | 44%  | 69%         |
> | Llama-3.2-1B-Instruct                 | 51%  | 73%         |
> | Llama-3.2-3B-Instruct                 | 90%  | 70%         |
> | Meta-Llama-3-8B-Instruct              | 85%  | 64%         |
> | Llama-3.1-70B-Instruct                | 82%  | 92%         |
> | Mixtral-8x7B-Instruct                 | 66%  | 70%         |
> | Qwen2.5-1.5B-Instruct                 | 53%  | 66.7%         |
> | DeepSeek-R1-Distill-Qwen-1.5B         | 33.7%  | 53.8%         |
> | Qwen2.5-7B-Instruct                   | 63%  | 77%         |
> | DeepSeek-R1-Distill-Qwen-7B           | 64.6%  | 78.3%         |
> | Qwen3-4B                              | 58%  | 51%         |
> | Qwen3-8B                              | 49%  | 50.1%         |
> | Qwen3-32B                             | 52%  | 55%         |
>
> We find that in many cases (even all models on ComplexBench) the base model achieves a win rate above 50%, indicating that it produces higher-quality responses than the reasoning variant, even when both follow instructions perfectly. This result suggests that reasoning can not only hurt instruction following, but may also degrade response quality in tasks where step-by-step reasoning is unnecessary.
>
>
> # 3. More Related Work
> These papers are indeed relevant and very interesting, and we are excited to see these related works come out recently. We have included them into the related work section of our paper. We really thank the reviewer for mentioning these papers! The **exploration of CoT length is particularly intriguing, and we also conducted some preliminary experiments** out of curiosity. In Appendix I, we already examined the correlation between the accuracy difference of the Base and Reason models and the length of the reasoning tokens, and observed no clear correlation. Below, we **conduct additional experiments to study the correlation between the accuracy difference and the length difference** between the Base and Reason models.
>
> More precisely, we conducted this analysis and report the Pearson correlation between accuracy difference (*Acc(Base) – Acc(Reason)*) and token length difference (*Len(Reason) – Len(Base)*) for both IFEval and ComplexBench. Overall, we do not observe strong correlations, except for DeepSeekR1-Distill-Qwen7B on IFEval. We omit p-values for brevity, but note that this case had a p-value of 2e-8, the only one below 5%. In **this particular case, it suggests that longer reasoning may be significantly associated with a larger drop in instruction-following performance**. We will include these results and further discussion in Appendix I.
>
> | Model                         | IFEval   | ComplexBench |
> |------------------------------|----------|---------------|
> | GPT-4o-mini                  | -0.047   | 0.066         |
> | Claude-3.5-haiku             | 0.068    | -0.015        |
> | Claude-3.5-sonnet            | -0.038   | -0.119        |
> | Claude-3.7-sonnet            | -0.119   | -0.035        |
> | DeepSeek-V3                  | -0.046   | -0.069        |
> | Llama-3.2-1B                 | 0.037    | 0.079         |
> | Llama-3.2-3B                 | 0.008    | -0.020        |
> | Llama-3.8B                   | 0.028    | 0.028         |
> | Llama-3.1-70B                | 0.074    | 0.022         |
> | Mixtral-8x7B                 | 0.090    | 0.002         |
> | Qwen2.5-1.5B                 | 0.073    | -0.137        |
> | DeepSeekR1-Distill-Qwen1.5B  | 0.136    | -0.045        |
> | Qwen2.5-7B                   | 0.137    | 0.017         |
> | DeepSeekR1-Distill-Qwen7B    | 0.238    | 0.035         |
>
>
> # 4. Other Mechanistic Interpretability Methods
> We agree that additional MI tools can illuminate the phenomena in our paper. In fact we are actively exploring some other ideas:
> * **Activation probing as a reasoning gate.** A simple linear probe trained on mid-layer activations might be able to predict whether CoT will help.  When the probe signals “reasoning unnecessary,” the model stays in Base mode. Otherwise it switches to CoT, reducing harmful reasoning.
> * **Logit-lens comparison.** Running Base and Reason side-by-side, we can project each layer’s residual stream through the LM head (logit lens) and locate constraint-relevant prompt positions where their vocabulary distributions diverge the most.  Then inspecting the hidden-state difference at these hotspots should reveal the essential causes of their different performances.
> * **Activation steering to mitigate the drop.** To further mitigate the drop (i.e. improve instruction following ability), another method we could think of is to perform activation steering (pre-compute the steering vectors for different types of constraints, or they could be the hidden-state difference identified above).  Injecting a small multiple of this vector at generation time could nudge outputs back toward instruction compliance without changing model weights.
>
> # 5. Combination of Multiple Strategies
> We agree these methods are relatively orthogonal and could be combined, although with potentially increased complexity. In this paper, our main goal is to systematically reveal and analyze the drop in instruction-following caused by reasoning/CoT through controlled experiments, case studies, and attention analysis. Thus, we focused on straightforward mitigation methods that do not alter model weights, since weight changes may negatively affect performance on other tasks. For instance, we **conducted additional experiments exploring the strategy of directly training the model for instruction following**. We do SFT on an IFEval subset and found that it **improved instruction-following for some models but reduced performance on general and reasoning benchmarks** (detailed results can be found in “2. Instruction-Following Training” in our response to Reviewer TQ5w below). Further combining multiple mitigation strategies and optimizing their interactions is an exciting direction that we plan to explore in future work!
>
> Overall, we deeply appreciate the reviewer’s insightful comments and valuable suggestions. We sincerely hope our clarification and additional experiments above would address the concerns, and respectfully hope the reviewer could further raise the score.

---

> > ### Author Response · Authors · 2025-08-04
> > **Reply to Reviewer SzjC**
> >
> > (Note: The response from Reviewer SzjC appears below in the thread with Reviewer TQ5w. We are quoting and responding here in the original thread for the convenience of the Area Chairs and Program Chairs.)
> >
> > Reviewer SzjC’s Response (04 Aug): *“Thanks to the authors for their elaborated response. I will keep my original score, suggesting to accept the paper to the conference.”*
> >
> > We sincerely thank the reviewer for the positive response and recommendation for acceptance! We are especially grateful for the insightful suggestions regarding multilingual evaluation, broader task generalization, related work on CoT length, and alternative mechanistic interpretability methods. All these comments significantly helped us improve our analysis and expand the scope of the discussion. We truly appreciate the reviewer’s time, thoughtful feedback, and engagement throughout the review process.

---

### Official Review · Reviewer_Rspa · 2025-07-01

**Clarity:** 3
**Significance:** 4
**Originality:** 2
**Rating:** 5
**Confidence:** 4

**Summary:**

This paper investigates the performance large language and reasoning models with respect to instruction following under chain-of-thought prompting. Experiments have been conducted with different models (both open and proprietary) of varying capabilities. The authors find that instruction following degrades under COT conditions for almost all models on both benchmarks. They then apply several mitigation methods to attempt to restore instruction following capabilities. The paper also proposes and explores constraint attention to examine the degradation of model performance for instruction following.

**Questions:**

-	Is the decreased performance in instruction following caused by the reasoning aspects of COT/reasoning models, or is it simply a byproduct of the increasing sequence length?
-	To follow up on that, in Appendix I, this question is partially answered by showing the length of the reasoning for the COT response is NOT strongly correlated to the difference in score between the COT and base models. This to me unintuitive, can we see correlations between the length of the tokens and the model accuracy as well as the differences in total token length between the COT and base models. Apologies if I missed this in the paper somewhere.
-	When COT prompting a reasoning model, do the models exhibit excessive reasoning where the over analysis may lead to failure? What is the difference in token count between the original reasoning model response and the COT reasoning model response and is that correlated with the performance?
-	Deepseek distill models had the most benefit from few-shot demonstrations, but their performance was also the weakest. Since Deepseek is a “reasoning” model it seems like they weren’t trained with extensive instruction following capabilities to begin with, and mainly distilled the reasoning ability of DeepSeek-R1.
-	I would like to know more about how the manual examination (L155) process was done, in addition to the resulting categories and the insight provided.
-	The choice of using 0 temperature (L128): I would like to see the reason for choosing this temperature.

**Ethical Concerns:**

["NO or VERY MINOR ethics concerns only"]

**Final Justification:**

The responses from the authors addressed most of my questions. I still have concerns about the discussion with regard to the previous work like [1], and simply saying "our study is complementary" is not satisfying. To reflect that most my concerns have been addressed, I increase my score accordingly.

**Limitations:**

yes

**Quality:**

3

**Strengths And Weaknesses:**

Strengths

- Many work pipelines require LLMs to adhere to instructions under COT, so it is important to study this topic.

- This paper runs experiments on a wide range of models which many different levels of capability.

- The paper makes some interesting new observations with regard to COT prompting for LLMs. They also provide several different methods for fixing these instruction following issues.

- The paper additionally proposed constraint attention and showed how it can be used to examine failure cases for instruction following.

- Overall, the paper is well-motivated and the flow of analysis and proposed methods is clear.

Weaknesses

- While the initial findings of COT harming instruction following is interesting, the follow-up mitigation methods are quite straightforward. Also, CoT may work due to different reasons. Investigation based the existing results (e.g., [1]) could help provide more insights.

- I believe there is a missing citation [2], on the effects of output constraints (e.g. word count, formatting) and context adherence (i.e. faithfulness to RAG facts) as two separate entities. In this paper, the authors collate the two and contrast them with CoT reasoning. It would strengthen the paper to add commentary on the distinction between these two types of instructions (output constraints vs. context adherence), especially since they observe in line 166-171 that CoT reasoning enhances some output constraints.

- There are no statistics to support the trends in lines 165-182 - some basic values such as average % increase in correct formatting, average % new errors in word count, etc. I believe this would improve the clarity/transparency of their observed trends. Since the authors point these out specifically, it would also help to understand if the mitigation strategies effectively reverse these trends, or if they are improving instruction following elsewhere.

- For method 4, lines 283-285, the authors mention they use a 50-50 split to train an external classifier. It is unclear if the 50-50 split is consistent across all experiments - if not, that is a significant drop in test samples (~750 compared to ~1500) and it is hard to say the results on Method 4 are comparable to the other experiments.

- Figure 1 and 2 use a WIN/LOSE metric to split CoT generations and there is some score metrics, but I did not see an explanation for the score.

- Overall, I would have liked more discussion/analysis about the individual models tested, even at the cost of fewer mitigation methods reported. While the thesis is clear, the analysis is too high-level considering the number of models tested.

[1] https://arxiv.org/pdf/2304.03843

[2] Wu, Z., Zhang, Y., Qi, P., Xu, Y., Han, R., Zhang, Y., Chen, J., Min, B., & Huang, Z. (2024). Dancing in Chains: Reconciling Instruction Following and Faithfulness in Language Models. Conference on Empirical Methods in Natural Language Processing. https://arxiv.org/pdf/2407.21417

---

> ### Author Rebuttal · Authors · 2025-07-29
>
> **TLDR:** *We clarify our choice of lightweight mitigation methods, incorporate suggested related works, and provide statistical analyses supporting our qualitative findings. We also address concerns about evaluation setups by adding Monte Carlo cross-validation, correlation analyses, and further experiments (e.g., CoT on Reason models). Additionally, we clarify figure interpretations and justify our choice of deterministic evaluation settings.*
>
> Thank you for the thoughtful and encouraging feedback. We’re especially grateful that you found our motivation, model-scale experiments, proposed methods, and constraint attention analysis to be clear and meaningful contributions. We also appreciate the insightful comments. Below are our responses:
>
> # 1. Straightforward Mitigation Methods
>
> We agree that the mitigation methods are straightforward. Our main focus is to reveal the drop in instruction-following caused by reasoning/CoT, supported by experiments, case studies, and attention analysis. For mitigation, we focus on light-weight methods that can be applied at inference time. We have limited API access to those proprietary LLMs and methods that involve changing model weights (e.g. specialized post-training recipes) may affect performance on other tasks. For example, we added SFT experiments on an IFEval subset: while some models improved in instruction-following, performance dropped on other general and reasoning tasks (please see results for “2. Instruction-Following Training” in our response to Reviewer TQ5w.)
>
> # 2. More Related Works
> We appreciate the reviewer for highlighting these relevant papers. [1] shows CoT helps link sparsely co-occurring information, and [2] examines the trade-off between instruction-following and faithfulness. Our study is complementary, and we will add the discussion of both in the Related Work section.
>
> # 3. Statistical Support for Case Study Summary
> We conducted a quantitative analysis of general constraint type distributions across models, comparing cases where reasoning helps vs. hurts. Below, we show **WIN/LOSE constraint-type distributions for two models** on IFEval. (Detailed distributions for all models/tasks will appear in Appendix.)
>
> **GPT-4o-mini**
>
> * *Reason WIN:* keywords 19.5%, length_constraints 17.1%, change_case 24.4%, combination 2.4%, detectable_format 19.5%, detectable_content 9.8%, punctuation 4.9%, startend 2.4%
>
> * *Reason LOSE:* combination 35.2%, length_constraints 31.0%, change_case 5.6%, keywords 16.9%, startend 2.8%, detectable_content 2.8%, punctuation 4.2%, detectable_format 1.4%
>
> **Claude-3.5-Haiku**
>
> * *Reason WIN*: change_case 30.0%, keywords 25.0%, length_constraints 15.0%, detectable_format 20.0%, detectable_content 5.0%, combination 5.0%
>
> * *Reason LOSE*: combination 43.3%, length_constraints 25.4%, keywords 10.4%, startend 4.5%, change_case 4.5%, detectable_content 1.5%, detectable_format 6.0%, language 3.0%, punctuation 1.5%
>
> We notice that for GPT-4o-mini,
>
> **Reasoning Helps ✓**: 81 % of the Reason‐wins come from format / lexical categories: detectable-format (19.5 %), change-case (24.4 %), keywords (19.5 %), length-constraints (17.1 %).
>
> **Reasoning Hurts ✗**: 66 % of the Reason‐losses are classic “over-focus / extra content” errors: combination-repeat-prompt & two-responses (35.2 %), length-constraints over-runs (31.0 %). This cleanly mirrors patterns 3 and 4 in our case study in Section 4.1.
>
> Claude-3.5-Haiku shows similar trends: lexical precision dominates wins, while losses are due to extra text/simple counting errors.”
>
> # 4. Different Evaluation Sample Size for Method 4
> We understand the concern regarding different test set sizes. To address this, we performed **Monte Carlo cross-validation by repeating random hold-out three times**. We also evaluated the latest Qwen3 models (which have thinking mode). Below, we report the **average**, **standard deviation**, **improvement** scores of the mitigation Method 4, and **p-values** from the two-sided z-tests:
>
> |Model| | |IFEval| | | |ComplexBench| |
> |-|-|-|-|-|-|-|-|-|
> ||AVG|STD|Improve|p-value|AVG|STD|Improve|p-value|
> |GPT-4o-mini|83.4|1.4|6.5|8.9e-16|66.0|0.8|5.7|5.5e-35|
> |Claude-3.5-Haiku|87.6|1.4|8.1|1.2e-23|68.9|0.7|6.8|1.6e-63|
> |Claude-3.5-Sonnet|87.8|0.3|8.3|0|67.9|0.2|1.9|7.8e-61|
> |Claude-3.7-Sonnet|90.6|0.2|0.4|0.00053|69.8|0.1|2.8|0|
> |Deepseek-V3|85.2|0.3|0.9|2e-07|71.8|0.1|0.7|7.8e-34|
> |Llama-3.2-1B-Instruct|53.1|1.2|12|1.2e-71|36.6|0.5|11|3.6e-295|
> |Llama-3.2-3B-Instruct|72.3|1.3|10|1.2e-43|50.7|0.6|5|3.2e-47|
> |Meta-Llama-3-8B-Instruct|76.3|1.3|17|1.5e-117|56.0|0.3|1.1|2.1e-10|
> |Llama-3.1-70B-Instruct|85.4|2.0|8.1|2.3e-12|68.5|0.7|8.3|1e-93|
> |Mixtral-8x7B-Instruct|54.1|0.5|-2.3|1.6e-15|60.9|0.3|2.6|6.2e-51|
> |Qwen2.5-1.5B-Instruct|37.1|0.8|5.5|1.1e-32|44.7|0.3|5.9|2.5e-254|
> |DeepSeek-R1-Distill-Qwen-1.5B|19.2|0.8|5.5|1.1e-32|19.1|0.1|2.4|0|
> |Qwen2.5-7B-Instruct|66.5|2.6|8.8|4.6e-09|63.4|0.5|11|2.5e-306|
> |DeepSeek-R1-Distill-Qwen-7B|32.1|1.3|7|1.1e-20|45.3|0.2|6.7|0|
> |Qwen3-4B|86.0|0.5|16|0|66.5|0.2|7.2|0|
> |Qwen3-8B|87.6|0.8|1.3|0.0049|67.6|0.2|7.8|0|
> |Qwen3-32B|88.8|0.9|3.8|2.6e-13|71.7|0.5|8.6|5.1e-195|
>
> We also computed the **overall p-value from a paired t-test** on improvements across all models: **6.3e-05 for IFEval and 2.1e-06** for ComplexBench. These significant improvements support the robustness and effectiveness of Method 4.
>
> # 5. Explanation of Figure Titles (Figure 1 and 2)
> We thank the reviewer for pointing this out. The accuracy evaluation for IFEval and ComplexBench is based on the number of instructions the model follows. The list of “True” and “False” in Figures represents boolean values indicating whether each response satisfies the constraints, and the “score” refers to the total count of “True” values. Thank you for pointing this out and we will clarify this in the paper.
>
> # 6. Correlation between Accuracy and Token Length Difference
> We agree that using token length difference may be a more natural choice. We conducted this analysis and reported the Pearson correlation between accuracy difference (*Acc(Base) – Acc(Reason)*) and token length difference (*Len(Reason) – Len(Base)*) for both IFEval and ComplexBench. Overall, we do not observe strong correlations, except for DeepSeekR1-Distill-Qwen7B on IFEval. We omit p-values for brevity, but note that this case had a p-value of 2e-8, the only one below 5%. In this particular case, it suggests that longer reasoning may be associated with a larger drop in instruction-following performance. We will include these results and further discussion in Appendix I.
> | Model|IFEval|ComplexBench|
> |-|-|-|
> |GPT-4o-mini|-0.047|0.066|
> |Claude-3.5-Haiku| 0.068 |-0.015|
> |Claude-3.5-Sonnet|-0.038|-0.119|
> |Claude-3.7-Sonnet|-0.119|-0.035|
> |DeepSeek-V3|-0.046|-0.069|
> |Llama-3.2-1B|0.037|0.079|
> |Llama-3.2-3B|0.008|-0.020|
> |Llama-3.8B|0.028|0.028|
> |Llama-3.1-70B|0.074|0.022|
> |Mixtral-8x7B|0.090|0.002|
> |Qwen2.5-1.5B|0.073|-0.137|
> |DeepSeekR1-Distill-Qwen1.5B|0.136|-0.045|
> |Qwen2.5-7B|0.137|0.017|
> |DeepSeekR1-Distill-Qwen7B|0.238|0.035|
>
> # 7. CoT Prompt a Reasoning Model
> We appreciate the suggestion. We **added experiments applying a CoT prompt to a reasoning model** (double thinking, denoted “ThinkCoT”) on three Qwen3 models. We observe that **it produces longer responses with two reasoning segments, and native thinking tends to be conversational, while prompted CoT more formal**. We observe modest gains of "ThinkCoT" over “Think” in some cases (e.g. 8B/32B on ComplexBench, 4B on IFEval), though still below “Base”. In other cases (e.g., 8B/32B on IFEval), performance drops. We do not see correlation to token length as longer “double-thinking” does not show consistent directional performance change over “Think” mode. In this situation, we suspect double thinking makes the model focus too much on thinking, affecting response quality.
> | Model|IFEval|ComplexBench|
> |-|-|-|
> |Qwen3-4B|85.0|63.8|
> |Qwen3-4B-Think|69.5|59.3|
> |Qwen3-4B-ThinkCoT|70.2|59.1|
> |Qwen3-8B|86.8|65.6|
> |Qwen3-8B-Think|86.3| 59.8|
> |Qwen3-8B-ThinkCoT|76.3|62.9|
> |Qwen3-32B|87.8|70.2|
> |Qwen3-32B-Think|85.0|63.1|
> |Qwen3-32B-ThinkCoT|75.8| 65.0|
>
> # 8. Manual Examination
> For each model, eight human annotators manually examined the responses of the Base and Reason variants. For each sample, we provided constraint-specific accuracy and asked annotators to write a short “TL;DR” summarizing why the Reason response won or lost. We found many recurring patterns in these summaries, which we categorized to support the insights presented in Section 4.1.
>
> # 9. Choice of Temperature
> For our instruction-following tasks and evaluation, we aim for more deterministic responses and scores. While using a nonzero temperature with multiple trials is an option, different models exhibit varying diversity at the same fixed temperature, and a limited number of trials may not accurately reflect true performance. Thus, we believe using temperature 0 is a more natural and reliable setting.
>
> We deeply appreciate the reviewer’s valuable comments. We hope our clarification and additional evaluations would address certain concerns, and sincerely hope the reviewer could raise the score.

---

> ### Author Response · Authors · 2025-08-06
>
> Dear Reviewer Rspa,
>
> Thank you again for taking the time to review our paper. We hope that our rebuttal has addressed your concerns and clarified our contributions. If you find our response helpful in resolving the issues raised, we would be sincerely grateful if you would consider updating your score to reflect your current assessment. If any further questions or uncertainties remain, we would be more than happy to further clarify during the discussion period!

---

> > ### Comment · Reviewer_Rspa · 2025-08-09
> >
> > The responses from the authors addressed most of my questions. I still have concerns about the discussion with regard to the previous work like [1], and the authors' simply reply, "our study is complementary", is not satisfying. To reflect that most my concerns have been addressed, I increase my score accordingly.

---

> ### Author Response · Authors · 2025-08-09
>
> We sincerely thank Reviewer Rspa for the kind reply. We are very glad to hear that our rebuttal helped address most of your concerns, and we truly appreciate your consideration in increasing the score!
>
> In fact, due to the length limit in the rebuttal response, we focused more on providing results to address other questions and kept our discussion in Section **"2. More Related Works"** brief. Below, we provide a more detailed response for this part:
>
> # 2.1 Connecting to Reference [1]
>
> Thank you for the suggestion to connect our findings with theoretical accounts of when CoT helps. Our paper analyzes *when*  and *why* CoT helps/hurts at the constraint level through case studies and a constraint-token attention analysis. Empirically, CoT degrades instruction following across models on IFEval and ComplexBench. Moreover, mechanistically, we find CoT often diverts focus away from simple constraints (e.g. exceeding length limits or inserting extra redundant text), while helping some format/lexical constraints.
>
> We will add an explicit discussion linking these observations to [1]. Their theory shows thinking steps help **when tasks require bridging non-local dependencies ("reasoning gaps")**. In our setting, many IFEval or ComplexBench constraints are *locally checkable* (such as token-level counts, formatting), so they rarely require such non-local composition. Therefore, the extra reasoning from CoT can reduce attention to constraint tokens, explaining the observed harm. We currently also have a hypothesis: **the more an instruction requires non-local composition (e.g. ComplexBench constraints with type Nested/Chain), the less harmful CoT should be**, which is consistent with [1]. We will include a small stratified analysis to further investigate this conjecture, while more detailed exploration is left for future work.
>
> # 2.2 Adding Citation [2]
>
> We agree and will cite [2]. That paper *empirically separates* two axes: **instruction following** and **faithfulness to provided context**. The authors show a trade-off when fine-tuning for one versus the other under a two-stage protocol.
>
> Our contribution is orthogonal and complementary: we study **how explicit CoT affects instruction following ability with rule-verifiable constraints** (e.g. word counts, formatting, lexical mentions) **without external context**. We demonstrated consistent CoT-induced drops and analyzed this phenomenon. As noted in our paper, CoT can *enhance* some output constraints (formatting/lexical precision), indicating that instruction following ability is multifaceted.
>
>
> # Planned changes
> - We will add a paragraph connecting our findings to [1]’s locality/reasoning-gap account and state the non-local-composition hypothesis, with preliminary explorations.
> -  We will also add a scoped paragraph explicitly distinguishing **output-format constraints** (our focus: IFEval/ComplexBench) from **context adherence/faithfulness** (RAG-style tasks in [2]). This positions our findings as *complementary* to the trade-off observed in [2].
> - Moreover, we also plan to include a brief note on applicability: our selective-reasoning methods naturally extends to RAG settings by adding a "context present" feature to the selector (i.e. the classifier). This predicts both whether the CoT would help this instruction and whether the context-faithfulness is at risk, which aligns with the observation in [2] that objectives can conflict. We will include this as a suggested extension in our revised paper.
>
> Finally, we are especially grateful for the insightful comments by Reviewer Rspa. Your feedback has led to valuable improvements in our work. We deeply appreciate your thoughtful evaluation and engagement throughout the discussion process.
>
>
> # References
>
> [1] Prystawski, A., et al. *Why think step by step?* (theoretical account: CoT helps when tasks exhibit non-local structure/reasoning gaps).
>
> [2] Wu, Z., et al. *Dancing in Chains: On Faithfulness and Instruction Following in Alignment.* (separates output-constraint following from context faithfulness; shows trade-offs under two-stage fine-tuning).

---

### Note · Authors · 2025-08-12

We sincerely thank the reviewers for their constructive feedback and the Area Chair for coordinating the discussion and reviewing our paper. Our central finding that explicit reasoning can harm instruction-following, was repeatedly recognized as novel, surprising, timely, and impactful. Reviewers praised the clarity of our motivation, the breadth of our model evaluation, the depth of our failure mode analysis via case studies and constraint attention, and the practicality of our mitigation strategies, noting the paper is well-written and addresses an underexplored but important problem.

**Reviewer Rspa:** Praised clear motivation, structured analysis, and meaningful contributions, calling our work "well-motivated" with a "clear flow of analysis and proposed methods." Requested stronger links to [1,2], more statistics for case studies, clarification of Method 4, and extra analyses (e.g., token length correlation, CoT on reasoning models). We added connection discussions, quantitative analyses, significance tests, clarified figures, and introduced new "double thinking" experiments.

**Reviewer SzjC:** Commended the thorough evaluation, novel constraint-attention analysis with visualizations, and concreteness of our best-practice recommendations. Suggested multilingual/generalization tests, non-reasoning tasks, related CoT-length work, and more interpretability methods. We clarified benchmark coverage, added LLM-judge quality evaluation, incorporated new related work, and discussed additional MI tools.

**Reviewer EvS7:** Described the finding as unexpected and important, and valued the in-depth qualitative failure mode analysis. Questioned Setting 1’s validity, sought significance tests, and broader quality evaluation. We justified both settings, added controlled reason/base comparisons, significance tests, and LLM-judge quality results.

**Reviewer TQ5w:** Highlighted our novelty, thoroughness, and effectiveness of our mitigations, and praised the writing quality. Asked about classifier generalization and direct IF training. We ran constraint-type holdout tests (showing strong generalization) and SFT experiments (noting trade-offs).

**Summary:** We are glad all reviewers noted our rebuttal addressed their concerns, keeping or changing to positive ratings. We appreciate the discussions, which reinforced our work’s clarity and further solidified its contribution to understanding and mitigating reasoning-induced instruction-following failures in LLMs.

---

### Decision · Program_Chairs · 2025-09-17

**Decision:**

Accept (spotlight)

**Comment:**

# Summary
This paper investigates a phenomenon where CoT prompting can significantly degrade instruction-following accuracy. They provided an analysis of when explicit reasoning can hurt/improve instruction-following and constraint satisfaction performance, and further compared four different mitigation strategies to restore the instruction-following capabilities. Specifically, they investigate the differences of these cases in terms of constraint attention trace, i.e., how the attention of output tokens to each constraint changes over decoding steps, on hundreds of samples. This study finds a major trend: reasoning flattens the constraint-attention trace, and the attention is lower on the answer segments when reasoning degrades the performance. The four mitigation strategies aim to judge and select reasoning vs. non-reasoning, and training an LLM-based classifier for this purpose outperforms others. All the empirical studies are conducted on IFEval and ComplexBench.

# Strengths
- Studying whether and when CoT can hurt/improve instruction following is an important, underexplored problem.
- The analysis provides novel insights into when CoT helps and where it should be used. It is novel to compare the differences made by CoT using constraint attention traces.
- The paper also compares different mitigation strategies and provides a feasible solution.
- Experiments are performed over a wide set of models and on two instruction-following benchmarks.
- The paper is well-motivated and well-written.

# Weaknesses
- The proposed mitigation strategies are quite straightforward and do not provide sufficient insights.
- Some important results are missing, e.g., the statistics of case studies, experiments on reasoning models, generalization of classifier-based strategy and its comparison to SFT baseline, the cost of mitigation methods, etc.
- The proposed method's generalizability to multilingual tasks and non-reasoning tasks is not evaluated.
- Insufficient discussion and comparison with works on CoT-length and more mechanistic interpretability.
- The evaluation setting needs more justification since the model may not be trained to perform the best under the CoT prompt.
- The analysis is limited to a few specific properties of the outputs when using or not using CoT.

# Reasons to Accept
- The studied problem is important for understanding when the widely-used CoT hurts/improves the instruction/constraint following capability.
- The mitigation strategies evaluated in this paper provide valuable insights to address the problem.
- The experiments cover various diverse models.
- The rebuttal and discussion addressed several major concerns of the reviewers with various additional experiments.

# Discussion Summary
- In the rebuttal, the authors provided detailed clarifications and a great amount of additional experiments to answer the questions of the reviewers.
- Most reviewers responded that the rebuttal and further discussion addressed their initial concerns.